# Retain and Adapt: Auto-Balanced Model Editing for Open-Vocabulary Object Detection under Domain Shifts

**Zixuan Duan, Fengyuan Lu, Xunzhi Xiang, Wenbin Li, Yang Gao, Qi Fan**✉
Nanjing University

## Abstract

Recent advances in Open Vocabulary Object Detection (OVOD) have shown strong performance on standard benchmarks, but performance drops sharply under out-of-distribution (OOD) shifts. Continual learning offers a potential remedy by sequentially integrating new tasks, yet existing methods often struggle to balance retaining the pre-trained model capabilities with adapting to new tasks, and usually require retraining under specific task orders. To address these limitations, we observe that model editing naturally lends itself to this setting, as it enables efficient knowledge injection while retaining prior capabilities. Building on this insight, we introduce **A**utomatically **B**alanced **M**odel **E**diting (**ABME**), which injects new task knowledge into the powerful OVOD models while preserving the model's original abilities. We first store compact key–value representations with storage cost independent of task volume. Then we leverage the stored KV matrices to automatically balance the new and old knowledge for varying learning scenarios, supporting order-agnostic task insertion or removal without additional retraining. Experiments show that ABME consistently achieves a better trade-off between maintaining pre-trained performance and adapting to diverse OOD tasks compared to existing continual learning approaches for OVOD, and generalizes seamlessly across different models and task scales. Project page is available here.

## 1 Introduction

OVOD has emerged as a promising paradigm that leverages vision–language pretraining to recognize a wide range of object categories without exhaustive annotations (Gu et al., 2021; Li et al., 2022b). Despite impressive progress on in-distribution benchmarks, recent studies reveal that OVOD models still struggle under distribution shifts, such as novel domains or previously unseen categories, leading to substantial performance degradation (Ilyas et al., 2024; Chhipa et al., 2024). In practice, domain shift can arise from changes in image style, capture conditions, or resolution, which significantly affect the reliability of detection even when the object categories remain the same. This limitation poses a critical challenge for deploying OVOD systems in real-world scenarios, where adaptability to diverse environments and continual integration of new knowledge are essential.

The conventional approach for continuously extending OVOD models to handle new concepts in multiple OOD domains is **continual learning**, which incrementally integrates new tasks while reducing forgetting across the sequence. Yet, they generally do not explicitly consider the pre-trained knowledge when balancing with the new capabilities required for multiple tasks, and they typically assume a fixed task order, limiting flexibility and efficiency; they are also highly order-sensitive, where small variations can yield large performance gaps (Lai et al., 2025; Hacohen & Tuytelaars, 2024). While **few shot fine-tuning** can perform well on a single few-shot dataset (Pan et al., 2025; Liu et al., 2025), it does not scale across multiple domains and frequently degrades base performance. These limitations motivate the search for a more lightweight and flexible alternative.

**Model editing** (Mitchell et al., 2021; Fang et al., 2024) has recently emerged as a lightweight approach for updating large pre-trained models, where small parameter modifications can effectively inject new knowledge without retraining the entire model. Importantly, it enables the insertion of

---

✉Corresponding author: fanqi@nju.edu.cn

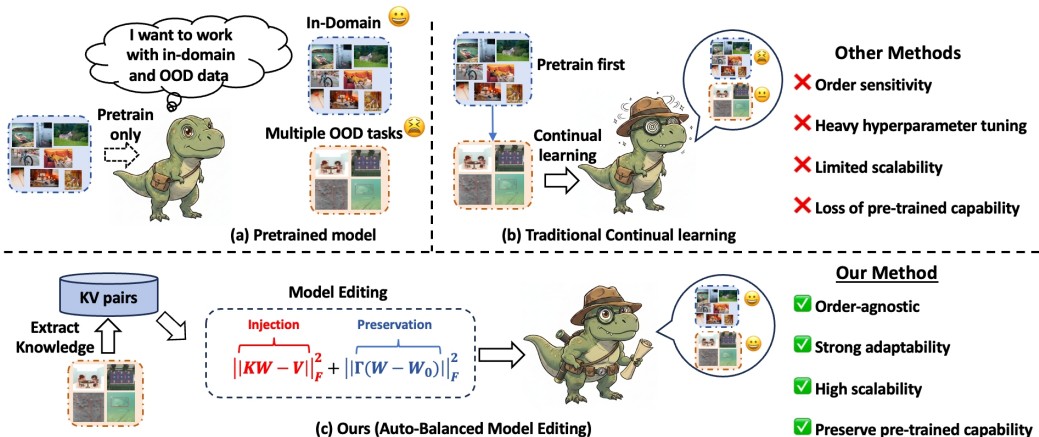

Figure 1: Comparison of paradigms for Open-Vocabulary Object Detection (OVOD) under domain shifts. (a) Pretrained models fail on OOD tasks. (b) Traditional continual learning adapts sequentially but suffers from order sensitivity, tuning overhead, and forgetting. (c) Our Auto-Balanced Model Editing (ABME) injects knowledge via key–value statistics, automatically balancing new and old tasks, achieving order-agnostic, scalable, and adaptive learning.

multiple new concepts while preserving the model's original capabilities, making it well-suited for continual extension in complex domains. In our OVOD few-shot experiments, we observe that fine-tuning only the Feed-Forward Network (FFN) parameters achieves performance comparable to full-model fine-tuning (see Table 11), suggesting that lightweight adaptation can be as effective as updating the entire model. Inspired by these experiments, we find that model editing provides a natural paradigm for continuously extending new concepts to OVOD models. However, while model editing has been predominantly explored in the context of large language models (LLMs), its adaptation to OVOD raises unique challenges that remain underexplored. As illustrated in Fig. 1, our approach contrasts with traditional continual learning by achieving order-agnostic, scalable, and adaptive knowledge integration.

To enable model editing in OVOD, we fine-tune only the FFN layers on the support set to obtain a task-adapted model, then pass the new-concept data through it to record the input (keys) and output (values) at the edited FFN layer as KV pairs. We store compact KV matrices independent of task number, ensuring scalability without extra memory cost. Yet injecting new knowledge through these KV matrices still requires tuning a hyperparameter to balance new and old knowledge across models and task scales, which is inefficient and impractical. To overcome this issue, we propose to use the KV matrices themselves to automatically adjust this balance, eliminating the need for tedious hyperparameter tuning and enabling a general solution applicable across diverse models and task volumes. Our approach offers several advantages: it ensures **Reliability**, effectively incorporating new knowledge and achieving up to 96.4% of fine-tuned performance on novel tasks; guarantees **Locality**, by preserving 94.1% of the pre-trained model's original capability; and provides strong **Flexibility**, as we can quickly perform direct combination or removal of task-specific KV statistics without retraining, making adaptation order-agnostic and scalable across diverse domains.

Our main contributions are as follows:

- **Introducing model editing to OVOD**: To the best of our knowledge, we are the first to bring model editing into the OVOD task, and we propose a principled way to construct key–value (KV) knowledge pairs for adapting new concepts.

- **Auto-balanced model editing**: We design a data-adaptive balancing strategy that automatically reconciles new and old knowledge, maintaining robustness across different models and task scales without manual trade-off tuning.

- **Extensive validation and generality**: We conduct comprehensive experiments on 19 few-shot datasets and validate our method on two distinct open-vocabulary detection models, Grounding DINO and GLIP, demonstrating its generality and scalability.

## 2 RELATED WORK

**Model editing.** Model editing aims to incorporate new knowledge while preserving performance on unrelated tasks, and existing approaches fall into two categories. Methods that *preserve the model's parameters* avoid modifying weights, instead relying on external mechanisms: IKE (Zheng et al., 2023a) and MemPrompt (Madaan et al., 2022) edit via prompts, SERAC (Meng et al., 2022b) introduces a counterfactual model, T-Patcher (Huang et al., 2023) adds correction neurons, and WISE (Wang et al., 2024) uses dual memory with conflict-free sharding. By contrast, approaches that *modify the model's parameters* directly update internal representations: Knowledge Neurons (Dai et al., 2022) alter selected neurons, MEND (Mitchell et al., 2021) employs a hypernetwork, ROME (Meng et al., 2022a) and GLAME (Zhang et al., 2024) use locate-then-edit strategies, AnyEdit (Jiang et al., 2025) performs autoregressive sequential editing, and AlphaEdit (Fang et al., 2024) constrains updates in the null space. Most of these methods focus on LLMs, where FFN layers are recognized as the primary locus of knowledge storage; our work extends this insight to OVOD, motivating an editing-based formulation for few-shot detection.

**Continual learning.** Continual learning addresses the challenge of catastrophic forgetting and can be grouped into three families: rehearsal-based, regularization-based, and architecture-based. *Rehearsal-based* approaches replay past knowledge using either stored exemplars or synthetic data, such as iCaRL (Rebuffi et al., 2017), CCL-GM (Lavda et al., 2018), GSS (Aljundi et al., 2019), and GDumb (Prabhu et al., 2020). *Regularization-based* methods constrain parameter updates to protect previously acquired information. Examples include BSS (Chen et al., 2019), EWC (Kirkpatrick et al., 2017), MAS (Aljundi et al., 2018), and diffusion-based strategies (Jha et al., 2024). *Architecture-based* methods expand or reconfigure the network to accommodate new tasks, e.g., DyTox (Douillard et al., 2022), LMC (Ostapenko et al., 2021), WSN (Kang et al., 2022), and Piggyback (Mallya et al., 2018). Although effective, these approaches are typically designed for long-term incremental learning and often require large-scale replay or architectural modifications. In contrast, our method targets few-shot OVOD and achieves knowledge retention efficiently by storing compact key–value statistics, without the need for exemplar memory or model expansion.

**OVOD.** OVOD has shown strong in-distribution results by leveraging vision–language pretraining and semantic alignment (Li et al., 2022b; Liu et al., 2024), yet recent studies show substantial degradation under distribution shifts with unseen categories or domains (Ilyas et al., 2024; Chhipa et al., 2024). To address this, several few-shot fine-tuning methods have been explored: ETS (Pan et al., 2025) adapts parameter configurations from foundation models, Domain-RAG (Li et al., 2025) retrieves semantically and stylistically similar images for data synthesis, and CD-ViTO (Fu et al., 2024) applies feature alignment, reweighting, and domain prompting. While effective on small datasets, these approaches are task-specific, requiring repeated fine-tuning that increases cost and may compromise prior performance.In contrast, we formulate few-shot adaptation as a model editing problem: instead of retraining, we inject task knowledge into FFN layers via compact KV statistics, enabling a single OVOD model to handle multiple tasks while retaining base-domain capability.

## 3 PRELIMINARY

### 3.1 MODEL EDITING IN LLMS

LLMs encode both factual and task-specific knowledge in their parameters, with recent studies showing that it is largely concentrated in the FFN layers (Meng et al., 2022b; Fang et al., 2024). Concretely, the FFN at the $l$-th layer can be written as:

$$\underbrace{m^l}_{v} = W_{\text{out}}^l \underbrace{\sigma\big(W_{\text{in}}^l \, \gamma(h^{l-1} + a^l)\big)}_{k}. \tag{1}$$

Here, $W_{\text{in}}^l$ and $W_{\text{out}}^l$ denote the input and output weight matrices of the FFN, respectively, while $\gamma$ represents layer normalization and $\sigma$ the activation function. The terms $a^l$ and $h^{l-1}$ correspond to the output of the attention block and the hidden state of the $(l-1)$-th layer. Prior work suggests that knowledge in LLMs can be abstracted as triples $(s, r, o)$, where $s$ denotes the subject, $r$ the relation, and $o$ the object (e.g., $s$ = The longest river in the world", $r$ = is", $o$ = "the Nile River"). In general, the intermediate representation before the output projection is taken as the key $k$, while

the projected output is defined as the value $v$, corresponding to $(s, r)$ and $(o)$, respectively, thereby forming key–value pairs (Geva et al., 2020). Building on this formulation, model editing seeks to replace original triples $(s, r, o)$ with updated ones $(s, r, o^*)$, where the new object $o^*$ is encoded through key–value pairs. Specifically, the new knowledge is represented by $K_1 \in \mathbb{R}^{d_0 \times u}$ and $V_1 \in \mathbb{R}^{d_1 \times u}$, which correspond to the key and value representations in the FFN output projection.

## 3.2 Model Editing Formulation for OVOD

We formalize the OVOD few-shot model editing problem as follows. Let $\mathcal{T} = \{\tau_1, \tau_2, \ldots, \tau_T\}$ be a set of $T$ tasks, where each task $\tau_i$ corresponds to an object detection problem sampled from datasets that are *OOD* with respect to the base OVOD model. In practice, these tasks often come from novel domains or contain unseen object categories, on which the pre-trained OVOD model typically performs poorly.

Each task $\tau_i$ is defined by a small support set $\mathcal{S}_i = \{(x_j, y_j)\}_{j=1}^{K_i}$, containing a few annotated examples, and a query set $\mathcal{Q}_i$ for evaluation. We treat the adaptation to task $\tau_i$ as injecting *new knowledge* into the model, analogous to model editing in large language models, where the model must quickly incorporate information from the support set to improve detection accuracy on the corresponding query set. The practical goal of OVOD knowledge injection is twofold: to achieve *Reliability*, the model should minimize the detection loss on the query sets $\mathcal{Q}_i$, i.e., $\mathcal{L}_{\text{det}}(\theta'; \mathcal{Q}_i)$; and to ensure *Locality*, it should simultaneously minimize the loss on the base dataset $\mathcal{D}_{\text{base}}$, i.e., $\mathcal{L}_{\text{det}}(\theta'; \mathcal{D}_{\text{base}})$. In this way, the model adapts effectively to new tasks while preserving its original capability. In our editing setting, only the support sets $\mathcal{S}_i$ are available for adaptation, while the query sets $\mathcal{Q}_i$ and the base dataset $\mathcal{D}_{\text{base}}$ are used solely for evaluation and are not accessible during editing. This setting is more practically relevant than conventional continual learning, which emphasizes long-term retention but overlooks preserving the model's original capability. These considerations motivate us to design an editing algorithm that efficiently injects new task knowledge while retaining the base model's capability, as described in Sec. 4.

## 4 Method

In this section, we first introduce the construction and storage of new knowledge key–value pairs (Sec. 4.1), then present the auto-balanced objective function (Sec. 4.2), and finally describe how to optimize this objective and summarize the overall procedure in Alg. 1 (Sec. 4.3).

## 4.1 Knowledge Key–Value Construction

Traditionally, knowledge editing in LLMs (Meng et al., 2022b) is formulated under the key–value framework. Given a key $k$ that encodes the subject–relation pair $(s, r)$, the model retrieves the corresponding value $v$ representing the object $o$. To inject new knowledge, existing methods usually keep the key $k$ fixed, while replacing the associated value $v$ with an updated value $v^*$ that encodes the target object $o^*$. In practice, $v^*$ is obtained by adjusting $v$ (e.g., via gradient descent) so that the model is more likely to predict $o^*$ when conditioned on $(s, r)$. To obtain the new knowledge key–value pairs in the OVOD model, we adopt a two-stage strategy consisting of **fine-tuning** and **storage**. In particular, we designate the parameter to be edited, such as the output projection matrix $W_{\text{out}} \in \mathbb{R}^{d_0 \times d_1}$ in the FFN Eq. 1, with its input and output defined as the key $k$ and value $v$, respectively. During fine-tuning, for task $\tau_i$, we regard the output projection matrix $W_{\text{out}}$ as the only learnable parameter while keeping all others frozen. The task-specific parameter set $\theta_i$ thus corresponds to the adapted $W_{\text{out}}$, which is optimized with the following traditional detection objective:

$$\min_{\theta_i} \mathcal{L}_{\text{det}}(\theta_i; \mathcal{S}_i), \tag{2}$$

Once the fine-tuned parameter $\theta_i$ is obtained, we feed the support set $\mathcal{S}_i$ into the corresponding fine-tuned OVOD model to extract the intermediate key–value representations, denoted as $K_i \in \mathbb{R}^{n_i \times d_0}$ and $V_i \in \mathbb{R}^{n_i \times d_1}$. However, the dimension $n_i$ can be extremely large; for instance, in a vision backbone it may scale with the number of samples and patches, making it prohibitive to store the full $K_i$ and $V_i$. In Sec. 4.3, we show that keeping the entire matrices is unnecessary. Instead, it suffices to store the covariance and cross-covariance matrices, i.e., $K_i^\top K_i$ and $K_i^\top V_i$, which are significantly more compact while preserving the information required for subsequent optimization.

---

**Algorithm 1** OVOD Auto-Balanced Model Editing (Batch-wise Accumulation)

---

**Input:** Pre-trained OVOD model (all parameters $\Theta$), chosen editable sub-parameter $W_0 \subset \Theta$ ($W_0 \in \mathbb{R}^{d_0 \times d_1}$), task datasets $\{\mathcal{S}_t\}_{t=1}^T$
**Output:** Updated editable weight $W^*$

1: $A \leftarrow 0_{d_0 \times d_0}, \quad B \leftarrow 0_{d_0 \times d_1}, \quad s \leftarrow \mathbf{0}_{d_0}$
2: **for** $t \in \{1, 2, \ldots, T\}$ **do**
3:     $\theta_t \leftarrow \arg\min\limits_{\theta \in W_0} L_{\det}(\theta; \mathcal{S}_t)$             $\triangleright$ Fine-tune only editable $W_0$, keep others in $\Theta$ fixed
4:     **for** each mini-batch $(x, y)$ from $\mathcal{S}_t$ **do**
5:         Forward with $\theta_t$ at the edited FFN layer to obtain $K_b \in \mathbb{R}^{n_b \times d_0}$, $V_b \in \mathbb{R}^{n_b \times d_1}$
6:         $A \leftarrow A + K_b^\top K_b$                             $\triangleright$ accumulate $K^\top K$
7:         $B \leftarrow B + K_b^\top V_b$                             $\triangleright$ accumulate $K^\top V$
8:         $s \leftarrow s + \left( \sum_{j=1}^{n_b} K_b[j,1]^2, \ldots, \sum_{j=1}^{n_b} K_b[j,d_0]^2 \right)$      $\triangleright$ accumulate $s_i$
9: $\Gamma^2 \leftarrow \mathrm{diag}(s^{1/2})$                                  $\triangleright \Gamma_{ii} = s_i^{1/4},\ \Gamma_{ii}^2 = s_i^{1/2}$
10: $H \leftarrow A + \Gamma^2$
11: $W^* \leftarrow H^{-1}\left(B + \Gamma^2 W_0\right)$                  $\triangleright$ solve via SPD linear system
12: **return** $W^*$

---

## 4.2 AUTO-BALANCED EDITING OBJECTIVE DESIGN

In order to effectively leverage the constructed key-value representations, we first introduce a straightforward objective: to allow the weight matrix $W$ to absorb new knowledge while retaining previously learned information, we consider the following simple formulation:

$$\min_W \ f(W) \ = \ \|KW - V\|_F^2 \ + \ \lambda \|W - W_0\|_F^2, \tag{3}$$

where $W_0$ denotes the original parameter of the OVOD model. The key–value matrices $(K, V)$ are obtained from the previous step by concatenating the knowledge matrices across all tasks:

$$K = \left[K_1; K_2; \ldots; K_T\right] \in \mathbb{R}^{n \times d_0}, \qquad V = \left[V_1; V_2; \ldots; V_T\right] \in \mathbb{R}^{n \times d_1}, \tag{4}$$

where the total number of samples is $n = \sum_{i=1}^T n_i$, with $n_i$ denoting the number of key–value pairs extracted from task $\tau_i$. The hyperparameter $\lambda$ controls the trade-off between fitting new knowledge and preserving the old one.

Through this optimization, the model is encouraged to acquire new knowledge while simultaneously preserving previously learned information. Apparently, the hyperparameter $\lambda$ cannot be derived analytically but must instead be set heuristically and subsequently evaluated on both the query set and the base set to determine its suitability. Moreover, the choice of $\lambda$ is not universal: it must be re-tuned whenever the task setting changes or a different OVOD model is employed.

A natural starting point is the $\lambda$-weighted $\ell_2$ regularizer in Eq. 3. However, $\lambda$ is task-dependent and requires re-tuning. To eliminate this hyperparameter, we design a data-adaptive reweighting optimization objective:

$$\min_W \ \|KW - V\|_F^2 \ + \ \|\Gamma(W - W_0)\|_F^2, \tag{5}$$

where $\Gamma \in \mathbb{R}^{d_0 \times d_0}$ is a positive diagonal matrix

$$\Gamma = \mathrm{diag}\left(s_1^{1/4}, \ s_2^{1/4}, \ \ldots, \ s_d^{1/4}\right), \quad s_i = \sum_t k_{ti}^2.$$

Intuitively, the regularization term $\|\Gamma(W - W_0)\|_F^2$ applies a weight of $s_i^{1/2}$ to the squared $\ell_2$ norm of the $i$-th row of $(W - W_0)$, since $\Gamma_{ii}^2 = s_i^{1/2}$. In this way, the objective balances new fitting information and prior knowledge in a data-adaptive manner, while a more detailed discussion of the benefits of this design is deferred to the Appendix B.3.

## 4.3 OPTIMIZATION AND OVERALL ALGORITHM

From Eq. 5, taking the derivative with respect to $W$ leads to a linear system. Let $H := K^\top K + \Gamma^2 \in \mathbb{R}^{d_0 \times d_0}$, which is symmetric and positive semidefinite in general, and becomes positive definite as long as $\Gamma$ has strictly positive diagonal. The unique minimizer is:

$$W^\star \;=\; H^{-1}\big(K^\top V + \Gamma^2 W_0\big). \tag{6}$$

In implementation, we never compute $H^{-1}$ explicitly; instead, we solve the linear system $HW^\star = K^\top V + \Gamma^2 W_0$ using a numerical solver (e.g., `torch.linalg.solve`). Moreover, computing the solution does not require storing the full key–value matrices. In practice, it suffices to maintain only the aggregated statistics $K^\top K$, $K^\top V$, and $\Gamma$, which can be obtained by summing the corresponding task-specific quantities:

$$K^\top K \;=\; \sum_{t=1}^{T} K_t^\top K_t, \qquad K^\top V \;=\; \sum_{t=1}^{T} K_t^\top V_t, \qquad \Gamma^2 \;=\; \sum_{t=1}^{T} \Gamma_t^2. \tag{7}$$

This additive update rule is similar in spirit to the matrix update in (Wang et al., 2021), and the detailed proof of equivalence is deferred to Appendix B.2. Although the above expressions are written as summations over all tasks from 1 to $T$ for convenience, they hold for any subset or combination of tasks within $\{1, \ldots, T\}$. Our method is summarized in Alg. 1.

## 5 EXPERIMENT

### 5.1 IMPLEMENTATION DETAILS

We conduct experiments on two cross-domain few-shot detection benchmarks, **CDFSOD** (Fu et al., 2024) and **ODinW-13** (Li et al., 2022a), covering 19 few-shot tasks under significant distribution shift. We follow the standard $K$-shot setting with $K \in \{1, 5, 10, 30, 50\}$. Further details of the datasets and training setups are provided in Appendix C.1.

**Evaluation metric.** Following CDFSOD and ODinW-13, we adopt the standard COCO-style Average Precision (AP) metric, computed over IoU thresholds from 0.50 to 0.95 across all object scales, for evaluation. In addition, we use the model's AP on the COCO dataset to measure its ability to retain the original detection performance.

### 5.2 MAIN RESULTS

To assess the trade-off between preserving prior knowledge and adapting to new tasks, we use two metrics: **Retention Ratio (RR)**, $\mathrm{AP}_{\mathrm{old}}^{\mathrm{edited}}/\mathrm{AP}_{\mathrm{old}}^{\mathrm{original}}$, which measures how well the model retains its original ability; and **Adaptation Gain Ratio (AGR)**, $\mathrm{AP}_{\mathrm{new}}^{\mathrm{edited}}/\mathrm{AP}_{\mathrm{new}}^{\mathrm{finetuned}}$, which evaluates adaptation to new tasks relative to full fine-tuning. Here, subscripts denote task type (old/new), and superscripts the model variant (original/edited/finetuned).

Tables 1, 2, and 3 summarize base performance, representative baselines: *EWC* (Kirkpatrick et al., 2017), *Adam-NSCL* (Wang et al., 2021), and *SD-LoRA* (Wu et al., 2025), as well as our editing method (*Ours*). Among them, Table 2 is our main experimental result. For reference, we also include *FFN fine-tune* on each target dataset as an oracle upper bound.

**Effectiveness on target tasks.** We report AGR to measure how effectively each method approaches full fine-

Table 1: Few-shot results on **ODinW-13**. Avg denotes the average performance across all 13 datasets.

| Shots | Method | Avg | COCO | RR | AGR |
|---|---|---|---|---|---|
| – | Base Model | 48.4 | 59.7 | – | – |
| 1-shot | EWC | 56.3 | 55.7 | 93.3% | 93.2% |
| | Adam-NSCL | 54.7 | 59.0 | 99.8% | 90.6% |
| | SD-LoRA | 55.2 | 54.4 | 91.1% | 90.1% |
| | Ours | 57.9 | 58.5 | 98.0% | 95.9% |
| | Oracle | 60.4 | – | – | – |
| 5-shot | EWC | 58.5 | 56.5 | 94.6% | 93.0% |
| | Adam-NSCL | 59.2 | 57.8 | 96.8% | 94.1% |
| | SD-LoRA | 58.4 | 55.6 | 93.1% | 92.8% |
| | Ours | 61.0 | 57.9 | 97.0% | 97.0% |
| | Oracle | 62.9 | – | – | – |
| 10-shot | EWC | 61.3 | 56.7 | 95.0% | 92.5% |
| | Adam-NSCL | 60.1 | 58.2 | 97.5% | 90.6% |
| | SD-LoRA | 61.1 | 55.8 | 93.5% | 92.2% |
| | Ours | 61.7 | 57.8 | 96.8% | 93.1% |
| | Oracle | 66.3 | – | – | – |

tuning. On CDFSOD-6, our method reaches *99%* (1-shot), *97%* (5-shot), *96%* (10-shot), *96%* (30-shot), and *95%* (50-shot), clearly higher than baseline methods. On ODinW-13, it achieves *96%* (1-shot), *97%* (5-shot), and *93%* (10-shot), again surpassing the baselines. These results indicate that our editing strategy retains near–fine-tuning adaptation while being more efficient than prior approaches.

Table 2: Few-shot results on **CDFSOD**. Avg denotes the average performance across all target datasets. "Oracle" refers to the results obtained by independently fine-tuning the FFN on each individual dataset, serving as an upper bound for performance.

| Shots | Method | ArTaxOr | Clipart1K | DeepFish | DIOR | NEU-DET | UODD | Avg | COCO | RR | AGR |
|---|---|---|---|---|---|---|---|---|---|---|---|
| – | Base Model | 12.8 | 49.1 | 28.6 | 4.5 | 1.2 | 10.1 | 17.7 | 59.7 | – | – |
| 1-shot | EWC | 9.6 | 53.5 | 30.6 | 10.8 | 7.4 | 8.2 | 20.0 | 57.6 | 96.5% | 90.9% |
| | Adam-NSCL | 12.4 | 51.5 | 32.1 | 5.8 | 1.4 | 6.9 | 18.4 | 59.0 | 98.8% | 83.6% |
| | SD-LoRA | 13.4 | 45.1 | 28.9 | 10.9 | 10.7 | 9.5 | 19.8 | 52.5 | 87.9% | 90.0% |
| | Ours | 12.1 | 54.8 | 33.6 | 10.5 | 5.0 | 14.1 | 21.7 | 57.5 | 96.3% | 98.6% |
| | Oracle | 13.6 | 55.8 | 31.9 | 11.8 | 5.6 | 13.5 | 22.0 | – | – | – |
| 5-shot | EWC | 33.3 | 54.4 | 35.7 | 19.5 | 20.0 | 23.4 | 31.0 | 57.1 | 95.6% | 78.3% |
| | Adam-NSCL | 68.2 | 52.1 | 33.8 | 8.4 | 5.0 | 7.0 | 29.1 | 57.8 | 96.8% | 73.5% |
| | SD-LoRA | 7.9 | 51.7 | 31.5 | 18.1 | 19.2 | 21.0 | 24.9 | 54.2 | 90.8% | 62.9% |
| | Ours | 69.4 | 57.9 | 37.6 | 26.1 | 20.2 | 19.5 | 38.5 | 57.0 | 95.5% | 97.2% |
| | Oracle | 69.2 | 60.2 | 35.1 | 29.4 | 22.2 | 21.5 | 39.6 | – | – | – |
| 10-shot | EWC | 21.3 | 56.5 | 39.9 | 24.4 | 23.3 | 25.7 | 31.9 | 55.5 | 93.0% | 74.4% |
| | Adam-NSCL | 72.2 | 52.8 | 34.6 | 11.4 | 10.0 | 11.7 | 32.1 | 58.2 | 97.5% | 74.8% |
| | SD-LoRA | 22.6 | 52.6 | 32.0 | 18.2 | 20.5 | 24.2 | 28.4 | 52.7 | 88.3% | 66.2% |
| | Ours | 71.6 | 59.5 | 38.7 | 32.4 | 22.8 | 20.9 | 41.0 | 56.8 | 95.1% | 95.6% |
| | Oracle | 72.3 | 61.4 | 36.9 | 37.1 | 25.0 | 24.4 | 42.9 | – | – | – |
| 30-shot | EWC | 35.1 | 54.9 | 38.1 | 23.1 | 18.3 | 27.6 | 32.9 | 54.1 | 90.6% | 67.3% |
| | Adam-NSCL | 75.1 | 53.5 | 37.5 | 13.2 | 8.7 | 7.6 | 32.6 | 57.9 | 97.0% | 66.7% |
| | SD-LoRA | 15.4 | 50.5 | 35.6 | 21.4 | 21.3 | 27.5 | 28.6 | 50.4 | 84.4% | 58.5% |
| | Ours | 76.5 | 58.1 | 39.9 | 45.2 | 32.2 | 26.9 | 46.7 | 55.1 | 92.3% | 95.5% |
| | Oracle | 77.8 | 61.1 | 44.3 | 49.7 | 35.1 | 25.6 | 48.9 | – | – | – |
| 50-shot | EWC | 46.2 | 52.0 | 36.0 | 23.8 | 16.6 | 27.1 | 33.6 | 52.8 | 88.4% | 66.5% |
| | Adam-NSCL | 79.7 | 52.9 | 34.9 | 20.5 | 12.3 | 13.9 | 35.7 | 57.2 | 95.8% | 70.7% |
| | SD-LoRA | 3.9 | 52.8 | 33.3 | 22.0 | 17.5 | 30.4 | 26.7 | 51.6 | 86.4% | 52.9% |
| | Ours | 79.5 | 58.6 | 35.4 | 51.2 | 34.5 | 28.9 | 48.0 | 54.5 | 91.3% | 95.0% |
| | Oracle | 81.9 | 62.4 | 40.2 | 54.0 | 35.9 | 28.4 | 50.5 | – | – | – |

**Preservation of prior capability.** We further evaluate the Retention Ratio (RR) to measure how much COCO performance is preserved after editing. Our method maintains a high RR across different shots (most exceeding 95%), indicating that even when achieving very high AGR on new tasks, the model still retains the vast majority of its original COCO capability. This demonstrates that our edits not only adapt effectively to novel domains but also avoid catastrophic forgetting on the source dataset.

**Flexibility.** An additional advantage of our approach lies in its flexibility. Without any extra training, we can directly combine the results from above and apply Eq. 7 to integrate the capabilities across all 19 tasks. As shown in Table 3, our method achieves high AGR (mostly above 95%) on new tasks while maintaining strong RR (also above 95%) on COCO, indicating that a *single edited model* can effectively adapt to all 19 datasets without retraining.

Table 3: Overall performance on all **19** datasets (CDFSOD-6 + ODinW-13). "Avg" denotes the average performance across all 19 datasets.

| Shots | Method | CDFSOD | ODinW | Avg | COCO | RR | AGR |
|---|---|---|---|---|---|---|---|
| – | Base Model | 17.7 | 48.4 | 38.7 | 59.7 | – | – |
| 1-shot | SD-LoRA | 18.2 | 58.1 | 45.5 | 53.4 | 89.4% | 94.2% |
| | Ours | 21.7 | 58.3 | 46.7 | 58.5 | 98.0% | 96.7% |
| | Oracle | 22.0 | 60.4 | 48.3 | – | – | – |
| 5-shot | SD-LoRA | 23.9 | 61.2 | 49.4 | 53.1 | 88.9% | 89.0% |
| | Ours | 37.3 | 60.2 | 53.0 | 58.0 | 97.2% | 95.5% |
| | Oracle | 39.6 | 62.9 | 55.5 | – | – | – |
| 10-shot | SD-LoRA | 25.0 | 61.4 | 49.9 | 51.8 | 86.8% | 84.7% |
| | Ours | 39.6 | 62.2 | 55.1 | 57.9 | 97.0% | 93.5% |
| | Oracle | 42.8 | 66.3 | 58.9 | – | – | – |

## 5.3 ABLATION STUDY

To evaluate generality under different task volumes, we gradually increase the number of CDFSOD target datasets by treating the first $k \in \{1, \ldots, 6\}$ as *new tasks* while keeping COCO as the old task. For the manual-tuning baseline, we use Eq. 3 with fixed $\lambda \in \{1, 5, 10, 15, 20\}$ and compute (i) COCO mAP for old-task retention and (ii) the mean mAP across the $k$ new tasks. We then average these results across all $k$ values to obtain an overall *old-task mean* and *new-task mean*. In contrast, our method directly applies the auto-balanced objective in Eq. 5 without any manual hyperparameters.

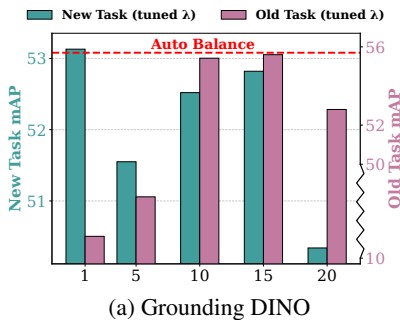

(a) Grounding DINO

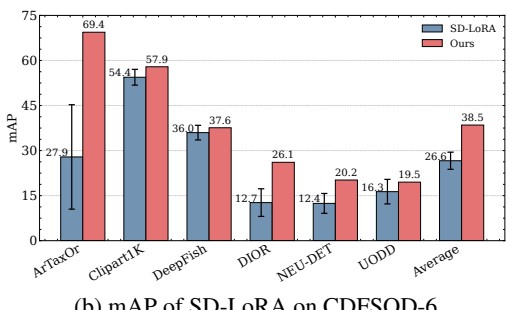

(b) mAP of SD-LoRA on CDFSOD-6

Figure 2: (a) Manual $\lambda$ tuning vs. Auto-Balanced Optimization, with the horizontal axis representing different values of $\lambda$, (b) mAP of continual learning methods on CDFSOD-6 across 20 random dataset orders. The error bars on SD-LoRA indicate the variance across different sequences.

Table 5: Continual learning methods combined with ours on **CDFSOD** 10-shot. This combination leads to significant improvements on the new CDFSOD tasks while also enhancing the retention of the original COCO task performance.

| Method | ArTaxOr | Clipart1K | DeepFish | DIOR | NEU-DET | UODD | Avg | COCO | RR | AGR |
|---|---|---|---|---|---|---|---|---|---|---|
| EWC | 21.3 | 56.5 | 39.9 | 24.4 | 23.3 | 25.7 | 31.9 | 55.5 | 93.0% | 74.4% |
| EWC+Ours | 73.3 | 59.3 | 39.6 | 31.6 | 23.8 | 22.3 | 41.7 | 55.7 | 93.3% | 97.2% |
| SD-LoRA | 22.6 | 52.6 | 32.0 | 18.2 | 20.5 | 24.2 | 28.4 | 52.7 | 88.3% | 66.2% |
| SD-LoRA+Ours | 69.5 | 57.9 | 34.8 | 26.4 | 22.0 | 20.1 | 38.5 | 56.0 | 93.8% | 89.7% |
| Oracle | 72.3 | 61.4 | 36.9 | 37.1 | 25.0 | 24.4 | 42.9 | – | – | – |

**Generality across task volumes.** Fig. 2a shows that the red dashed line (Auto-Balanced) consistently exceeds the bars corresponding to fixed-$\lambda$ settings in terms of new-task mAP, while simultaneously maintaining competitive COCO performance (right axis; compressed scale). This demonstrates that Auto-Balanced Optimization generalizes well across varying task volumes and outperforms careful manual tuning, all without the need for parameter search.

**Order-agnostic learning.** Fig. 2b evaluates continual learning methods on the six CDFSOD datasets by randomly shuffling their order and repeating the process 20 times. Results show that conventional continual learning is highly sensitive to task order, as indicated by the large error bars for SD-LoRA. In contrast, our method aggregates tasks according to Eq. 7, achieving stable performance regardless of task order.

As shown in Table 4, we conduct an ablation study on the exponent $\alpha$ in $\Gamma = \mathrm{diag}(s_1^\alpha, s_2^\alpha, \ldots, s_d^\alpha)$, where $s_i = \sum_t k_{ti}^2$. The results indicate that smaller exponents (e.g., $\frac{1}{4}$ or $\frac{1}{8}$) yield consistently better performance on CDFSOD-6 (5-shot) while maintaining comparable COCO accuracy.

**ABME combined with continual learning.** Table 5 shows that when applying our method for model editing on top of

Table 4: Ablation on exponent $\alpha$ in $\Gamma$ on **CDFSOD** 5-shot.

| $\alpha$ | CDFSOD | COCO |
|---|---|---|
| 1 | 0 | 0 |
| 1/2 | 36.6 | 57.0 |
| 1/4 | 38.5 | 57.0 |
| 1/8 | 38.5 | 56.9 |

conventional continual learning approaches, significant improvements are achieved. Under the CDFSOD 10-shot setting, EWC+Ours improves by 9.8 mAP, SD-LoRA+Ours improves by 10.1 mAP, and further gains are also observed on the COCO old tasks. Moreover, we trained the GLIP model on CDFSOD under the 1/5/10-shot settings using our method. As shown in Table 6, our method achieved an average RR of 97.9% and an AGR of 95.7% across the three settings, demonstrating its generalizability across different models.

**Ablation on Editing Regions.** Model editing in LLMs typically requires locating specific layers for knowledge injection. Following this, we screened FFN layers across different modules (e.g., backbone vs. encoder-decoder) Mitchell et al. (2021). As shown in Table 7, editing FFN layers across the entire architecture (Ours) achieves the best balance between new-task adaptation and old-task retention, outperforming edits restricted to specific sub-modules.

Table 6: GLIP few-shot results on **CDFSOD**. We apply our ABME method on GLIP under the 1, 5, and 10-shot settings. The results show that our approach achieves consistently high RR and AGR.

| Shots | Method | ArTaxOr | Clipart1K | DeepFish | DIOR | NEU-DET | UODD | Avg | COCO | RR | AGR |
|---|---|---|---|---|---|---|---|---|---|---|---|
| – | Base Model | 12.0 | 52.5 | 35.7 | 6.6 | 1.7 | 5.9 | 19.1 | 59.4 | – | – |
| 1-shot | Ours | 32.9 | 55.5 | 36.1 | 16.1 | 11.1 | 4.2 | 26.0 | 58.8 | 99.0% | 98.1% |
| | Oracle | 33.8 | 55.2 | 36.2 | 17.3 | 11.9 | 4.5 | 26.5 | – | – | – |
| 5-shot | Ours | 51.8 | 55.1 | 41.5 | 28.1 | 20.5 | 14.1 | 35.2 | 58.1 | 97.8% | 94.6% |
| | Oracle | 54.7 | 54.9 | 43.7 | 31.3 | 21.2 | 17.1 | 37.2 | – | – | – |
| 10-shot | Ours | 51.4 | 56.5 | 39.1 | 34.1 | 23.5 | 18.6 | 37.2 | 57.6 | 97.0% | 94.4% |
| | Oracle | 53.1 | 56.7 | 41.0 | 38.3 | 25.4 | 21.7 | 39.4 | – | – | – |

Table 7: Ablation on screening FFN layers in different modules (Grounding DINO, 5-shot CDF-SOD). We report New-task mAP and COCO mAP. "Back." refers to FFNs in the Backbone, and "Enc-Dec" to FFNs in the Encoder-Decoder.

| Edited FFNs | Back.(shallow) | Back.(mid) | Back.(last) | Back. (All) | Enc-Dec (All) | Back.(mid)+Enc-Dec | All FFNs (Ours) |
|---|---|---|---|---|---|---|---|
| New-task | 23.4 | 22.3 | 20.6 | 35.1 | 33.9 | 36.2 | **38.5** |
| Old-task | 56.5 | 59.6 | 59.3 | 58.3 | 56.5 | 58.4 | **57.0** |

**Impact of Editing Location (FFN vs. Self-Attention).** To validate our design choice of targeting FFN layers, we compare the performance of ABME when applied to FFN versus Self-Attention (SA) layers. As shown in Table 8, editing FFN layers yields a superior adaptation performance of 38.5 mAP on CDFSOD, outperforming SA editing (36.3 mAP) by 2.2 mAP. This confirms that FFN layers are the more effective site for injecting semantic knowledge in OVOD models. However, it is worth noting that our method still recovers 93.6% of the independent fine-tuning performance (Layer Oracle) even on SA layers. This demonstrates that while FFN is the optimal location, our auto-balanced algorithm generalizes robustly across different network structures.

Table 8: Comparison of editing FFN vs. SA layers on CDFSOD (5-shot). "Layer Oracle" denotes the upper bound achieved by independently fine-tuning the specific layer type on each dataset.

| Editing Target | CDFSOD | Layer Oracle | % of Oracle | COCO |
|---|---|---|---|---|
| Base Model | 17.7 | - | - | 59.7 |
| Self-Attention Layers | 36.3 | 38.8 | 93.6% | 57.8 |
| FFN out Layers | 38.5 | 39.6 | 97.2% | 57.0 |

## 5.4 SCALABILITY AND VERSATILITY ANALYSIS

**Scalability to large-scale heterogeneous domains.** To simulate extreme domain shifts and long-term adaptation, we extended CDFSOD to 90 sequential tasks using ImageCorruptions (Michaelis et al., 2019). We specifically selected the most severe intensity (level 5 of 5) to challenge the model limits. As shown in Table 9, ABME achieves 23.8 mAP, closely matching the Oracle (26.4 mAP) while retaining 96.3% of the original COCO performance.

**Generalization to CLIP-based Classification.** To verify cross-architecture generalization, we applied ABME to CLIP (Radford et al., 2021) (ViT-B/16) (Dosovitskiy, 2020) on 11 diverse datasets under a 5-shot setting. As shown in Table 10, our unified model achieves 74.1% accuracy, outperforming specific continual learning baselines (ZSCL (Zheng et al., 2023b), WiSE-FT (Wortsman et al., 2022)) and the Zero-shot Base (65.3%). It recovers ∼92% of the Oracle(FFN out) performance (80.4%), demonstrating effective adaptation beyond object detection. Oracle (FFN-out) and Oracle (Full) refer to independent fine-tuning per dataset on FFN output layers and all parameters, respectively. Specific dataset details are provided in the Appendix C.1.

**Visualization results.** As shown in Fig. 3 and Fig. 4, the Base model (Grounding DINO pretrained) struggles under domain shifts, resulting in many missed detections, false positives, and heavily overlapping features. EWC and Adam-NSCL provide limited improvements in detection and feature separation, but still fail to generalize well. In contrast, our proposed ABME method achieves stronger detection of novel objects, reduces false detections, and produces more compact and clearly separated feature clusters, demonstrating superior robustness against domain shifts.

Table 9: Performance on 90 sequential tasks with extreme domain shifts (15 corruption types). Detailed average performance across the 15 corruption types is provided in Table 12.

| Method | 90 Tasks | COCO |
|---|---|---|
| Base Model | 10.1 | 59.7 |
| Adam-NSCL | 20.8 | 57.2 |
| Ours | 23.8 | 57.5 |
| Oracle | 26.4 | - |

Table 10: Generalization to CLIP on 11 classification datasets (5-shot).

| Method | Avg Acc (%) |
|---|---|
| Base (Zero-shot) | 65.3 |
| ZSCL | 67.4 |
| WiSE-FT | 71.9 |
| Ours | 74.1 |
| Oracle(FFN out) | 80.4 |
| Oracle(Full) | 77.5 |

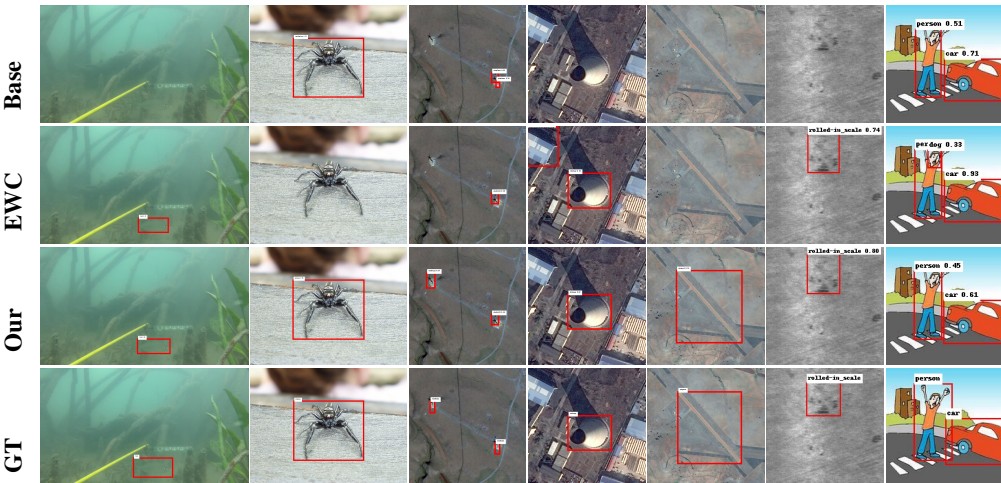

Figure 3: Comparison of Grounding DINO pretrained model (Base), EWC, our proposed method (Ours), and ground truth (GT) across different datasets.

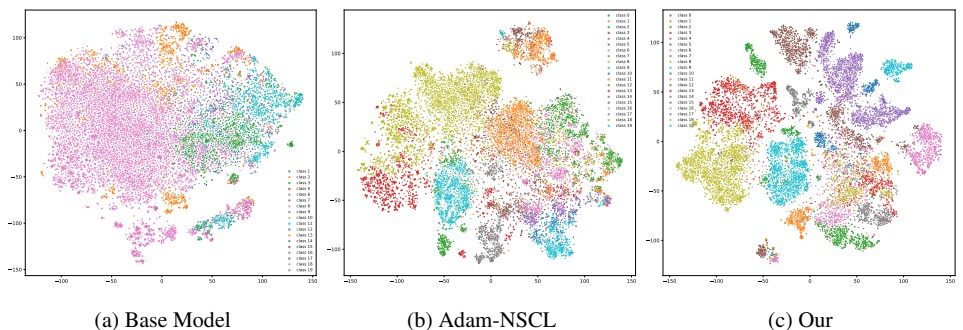

| (a) Base Model | (b) Adam-NSCL | (c) Our |
|---|---|---|

Figure 4: t-SNE visualization results of the Base model (Grounding DINO pretrained), Adam-NSCL, and our ABME method on the DIOR dataset, all trained on CDFSOD with 10-shot setting.

# 6 CONCLUSION

We proposed an auto-balanced model editing framework for OVOD that treats few-shot adaptation as knowledge injection into FFN layers and introduces a method that eliminates the need for tedious hyperparameter tuning by automatically balancing new and old knowledge. Our method achieves performance close to full fine-tuning on new tasks while retaining most original capabilities, and its compact key–value design supports flexible task combinations without retraining. We believe this editing-based perspective offers a lightweight and scalable alternative to traditional continual learning pipelines, and we hope it can inspire a new paradigm of continual learning for visual tasks, ultimately enabling more robust and adaptive open-world perception systems.

ACKNOWLEDGEMENTS

This work is supported in part by the National Natural Science Foundation of China (62192783, 62276128, 62406140), Young Elite Scientists Sponsorship Program by China Association for Science and Technology (2023QNRC001), the Key Research and Development Program of Jiangsu Province under Grant (BE2023019) and Jiangsu Natural Science Foundation under Grant (BK20221441, BK20241200). The authors would like to thank Huawei Ascend Cloud Ecological Development Project for the support of Ascend 910 processors.

ETHICS STATEMENT

All authors have read and agree to abide by the ICLR Code of Ethics. This work does not involve interventions with human participants or personally identifiable information. We use only publicly available datasets under their original licenses and follow the terms of use. Potential risks and our mitigations are summarized below:

- **Privacy & Security.** We do not collect or release any personal data. When showing qualitative examples, all images/videos are from public datasets; any sensitive content is filtered.
- **Bias & Fairness.** We report results on multiple benchmarks and provide detailed settings to facilitate external auditing. We acknowledge possible dataset biases and encourage follow-up evaluation on broader demographics and domains.
- **Dual Use / Misuse.** The method could be misused to enable undesired large-scale labeling or surveillance. To reduce misuse, we release only research artifacts (code/configs) and exclude any tools for scraping or re-identifying individuals.
- **Legal Compliance.** We comply with licenses of all third-party assets (code, models, and datasets) and cite their sources. Any additional third-party terms are respected.
- **Research Integrity.** We document preprocessing, training recipes, and evaluation protocols; random seeds and hyperparameters are provided to enable reproducibility.

Where applicable, institutional review information is withheld for double-blind review and can be provided after acceptance.

REPRODUCIBILITY STATEMENT

We provide detailed training configurations, including hyperparameters and optimization settings, in the main paper and Appendix. In addition, we will release code to ensure reproducibility, covering: (i) random seeds; (ii) full data preprocessing and splits; (iii) code structure with scripts to reproduce the main tables and figures; (iv) checkpoints and logs for the primary models.

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

## A  THE USE OF LLMS

We used ChatGPT-4o to polish our manuscript, using the following prompt:

```
I want you to act as an expert in scientific writing.  I will
provide you with some paragraphs in English and your task is
to improve the spelling, grammar, clarity, conciseness, and
overall readability of the text provided, while breaking down long
sentences, reducing repetition and increasing logic.  You should use
artificial intelligence tools, such as natural language processing,
rhetorical knowledge, and your expertise in effective scientific
writing techniques to reply.  Provide the output as a table in
a readable mode.  The first column is the original sentence, the
second column is the sentence after editing, and the third column
provides explanation of your edits and reasons.  Please edit the
following text in a scientific tone:
```

## B  THEORETICAL ANALYSIS

### B.1  ANALYSIS OF OPTIMIZATION FUNCTIONS

The optimization problem is given by:

$$\min_{W} \|KW - V\|_F^2 + \|\Gamma(W - W_0)\|_F^2 \tag{8}$$

where $\Gamma$ is a diagonal matrix defined as:

$$\Gamma = \mathrm{diag}(s_1^{1/4}, s_2^{1/4}, \ldots, s_d^{1/4}), \quad s_i = \sum_j k_{ij}^2 \tag{9}$$

To find the optimal solution, we take the first-order derivative and set it equal to zero:

$$K^T(KW - V) + \Gamma^2(W - W_0) = 0 \tag{10}$$

$$\Rightarrow (K^T K + \Gamma^2)(W - W_0) = K^T V - K^T K W_0 \tag{11}$$

$$\Rightarrow \Delta W = (K^T K + \Gamma^2)^{-1} K^T (V - K W_0) \tag{12}$$

### B.2  MATRIX UPDATING RULE

We have the following expressions for $K$ and $V$:

$$K = \begin{bmatrix} K_1 \\ K_2 \\ \vdots \\ K_T \end{bmatrix}, \quad V = \begin{bmatrix} V_1 \\ V_2 \\ \vdots \\ V_T \end{bmatrix}$$

For $K^T K$, we can express it as:

$$K^T K = \sum_{t=1}^{T} K_t^T K_t \tag{13}$$

For $K^T V$, we have:

$$K^T V = \sum_{t=1}^{T} K_t^T V_t \qquad (14)$$

### B.3 THEORETICAL ANALYSIS OF THE AUTO-BALANCING MECHANISM

The objective function is formally defined as:

$$\min_{W} \|(KW - V)\|_F^2 + \|\Gamma(W - W_0)\|_F^2, \qquad (15)$$

where the diagonal matrix $\Gamma$ is set as $\Gamma_{ii} = s_i^{1/4}$, with $s_i = \sum_t k_{ti}^2$ representing the accumulated feature energy of the input key matrix $K$. In practice, the raw feature energy $s_i$ inherently exhibits extreme variations across different feature dimensions and tasks (e.g., varying from $10^4$ to $10^6$ in our CDFSOD 5-shot experiments). This creates a severe scale imbalance for optimization: a fixed scalar $\lambda$ would be either too negligible for high-energy dimensions or too restrictive for low-energy ones. To resolve this, we derived our $\Gamma$ design based on the optimization curvature to automatically normalize these scales.

**Feature-wise Adaptivity via Curvature Analysis.** We address the optimization imbalance by analyzing the Hessian (curvature) of the objective function $\mathcal{L}$.

- **Data Term Curvature:** The Hessian of the first term $\|KW - V\|_F^2$ with respect to $W$ is $2K^T K$. Suppose the input feature $K$ has a magnitude scale $M_k$. The curvature of this term scales quadratically, i.e., $\mathcal{O}(M_k^2)$. This creates extremely steep optimization landscapes for high-energy features, leading to dominance in updates if not balanced.
- **Adaptive Regularization:** The Hessian of our regularization term is $2\Gamma^2$. Since the feature energy $s_i$ scales with $\mathcal{O}(M_k^2)$, our design $\Gamma_{ii} = s_i^{1/4}$ implies that $\Gamma^2$ scales linearly with $\mathcal{O}(M_k)$ (as $\Gamma^2 \propto \sqrt{s_i} \propto M_k$).

This formulation achieves a critical balance mechanism:

- **Adaptivity:** The regularization strength $\Gamma^2$ grows with feature magnitude ($\mathcal{O}(M_k)$), providing necessary constraints for strong features compared to fixed scalars.
- **Plasticity:** The regularization grows slower than the data term ($\mathcal{O}(M_k)$ vs. $\mathcal{O}(M_k^2)$). This "sub-quadratic" scaling prevents the regularization from becoming over-rigid (which would happen if $\Gamma_{ii} = s_i^{1/2}$, where $\Gamma^2 \sim \mathcal{O}(M_k^2)$), thereby allowing sufficient injection of new knowledge even for high-signal features.

## C EXPERIMENT

In our experiments, we primarily adopt **Grounding DINO** (Liu et al., 2024), a state-of-the-art open-vocabulary object detector that unifies grounding with strong detection performance. We also evaluate on **GLIP** (Li et al., 2022b), a vision-language pre-training model that aligns detection with phrase grounding, in order to verify the generality of our auto-balance strategy across different models and task scales.

For fine-tuning, we update only the output layers of all FFN modules. The batch size is set to 8 for Grounding DINO and 2 for GLIP, and we use the AdamW optimizer with a learning rate of $1 \times 10^{-4}$. Training is conducted for 18 epochs on most datasets, with slight adjustments under different shot settings to achieve better adaptation.

### C.1 TRAINING DETAILS

**CDFSOD.** The Cross-Domain Few-Shot Object Detection (CDFSOD) benchmark (Fu et al., 2024) uses MS-COCO (Lin et al., 2014) as the source domain and includes six heterogeneous target domains: ArTaxOr, Clipart1K, DIOR, DeepFish, NEU-DET, and UODD. These domains differ substantially in visual style, resolution, and object distribution, providing a challenging setup for cross-domain transfer.

Table 11: Few-shot results on **CDFSOD** 5-shot using **Grounding DINO-B**. The base model is the pre-trained Grounding DINO-B without fine-tuning. Methods marked with [†] denote the final model obtained by sequential fine-tuning on a single model, evaluated across all datasets.

| Method | ArTaxOr | Clipart1K | DeepFish | DIOR | NEU-DET | UODD | Average | COCO |
|---|---|---|---|---|---|---|---|---|
| Base model | 12.8 | 49.1 | 28.6 | 4.5 | 1.2 | 10.1 | 17.7 | 59.7 |
| FFN $W_{out}$ only | 69.2 | 60.2 | 35.1 | 29.4 | 22.2 | 21.5 | 39.6 | – |
| Fully fine-tune | 70.3 | 59.3 | 37.0 | 29.4 | 22.3 | 24.3 | 40.5 | – |
| Fully fine-tune[†] | 52.4 | 51.1 | 35.2 | 22.6 | 19.8 | 22.1 | 33.9 | 47.9 |

**ODinW-13.** ODinW-13 (Li et al., 2022a) is a subset of the ELEVATER benchmark (Li et al., 2022a), consisting of 13 diverse object detection tasks drawn from various open-world datasets. The benchmark emphasizes robustness to domain shifts and serves as a standard evaluation for open-vocabulary detectors.

**Few-shot setting.** Together, CDFSOD and ODinW-13 form 19 few-shot tasks with clear distribution shifts, on which open-vocabulary detectors typically experience performance degradation resembling out-of-distribution scenarios. We follow the standard $K$-shot protocol with $K \in \{1, 5, 10, 30, 50\}$, where $K$ labeled images per class are sampled as the support set and the remaining images are used for evaluation. For fair comparison, all baseline methods are consistently applied to the output layers across all FFN modules of the model.

**EWC.** For Elastic Weight Consolidation (EWC) (Kirkpatrick et al., 2017), we introduce a quadratic penalty on parameter updates, weighted by the Fisher information estimated from the previous task. The regularization coefficient is set to $\lambda_{\text{EWC}} = 10000$. We use the AdamW optimizer with a learning rate of $1 \times 10^{-4}$ and a batch size of 8, training for 18 epochs on most tasks. The overall objective can be written as

$$\mathcal{L}(\theta) = \mathcal{L}_{T+1}(\theta) + \frac{\lambda_{\text{EWC}}}{2} \sum_t \sum_i F_{t,i} \left( \theta_i - \theta_{t,i}^* \right)^2, \tag{16}$$

where $\mathcal{L}_{T+1}(\theta)$ denotes the standard loss on the current task, $F_{t,i}$ represents the Fisher information of parameter $\theta_i$ estimated from task $t$, and $\theta_{t,i}^*$ is the optimal parameter value obtained after training on task $t$. This regularization term penalizes large deviations from previously important parameters, thereby mitigating catastrophic forgetting across sequential tasks.

**Adam-NSCL.** For Adam-NSCL (Null-Space Continual Learning with Adam) (Wang et al., 2021), we constrain parameter updates to the approximate null space of previously learned features, following the SVD-based formulation. Specifically, we select the null-space basis $U_2^l$ associated with the smallest singular values of $\Lambda_2^l$, and adaptively choose $\Lambda_2^l$ with diagonal entries $\lambda \in \{\lambda \mid \lambda \leq a\lambda_{\min}^l\}$, where $\lambda_{\min}^l$ is the smallest singular value. In our experiments, we set the hyperparameter $a = 40$. Optimization is performed using the Adam optimizer with a learning rate of $1 \times 10^{-4}$, a batch size of 8, and training for 18 epochs on most tasks.

**SD-LoRA.** For SD-LoRA (Spectral Decay LoRA) (Wu et al., 2025), we adopt the same adapter structure as LoRA but additionally apply spectral decay regularization on the low-rank updates to improve stability across sequential tasks. The hyperparameters are set as follows: rank $r = 16$, initial spectral decay coefficient $\alpha_{\text{init}} = 1.0$, dropout rate 0.0, LoRA learning rate 0.002, and LoRA weight decay 0.0. Training is performed for 18 epochs on most tasks with a batch size of 8.

**Training Details for CLIP Experiments.** We evaluated our method on 11 diverse classification datasets, organized in the following order: Aircraft, Caltech101, CIFAR100, DTD, EuroSAT, Flowers102, Food101, MNIST, OxfordPets, StanfordCars, and SUN397. For optimization, we standardized the training duration to 500 iterations per dataset. Regarding the scope of parameter updates, our ABME method specifically targets the output projection layers of all FFN blocks within the ViT-B/16 (Dosovitskiy, 2020) encoder, whereas the baseline methods (ZSCL (Zheng et al., 2023b) and WiSE-FT (Wortsman et al., 2022)) involve full-model fine-tuning.

Table 12: Performance comparison of different methods under various corruptions on CDFSOD 5-shot. All values represent mean Average Precision (AP) in percentage. Note that **Oracle** is obtained by independent fine-tuning on each corrupted dataset (averaged over 90 total datasets). Continual learning setting follows a sequential training protocol according to the table rows; specifically, within each corruption, it trains on ArTaxor, Clipart1k, DeepFish, DIOR, NEU-DET, and UODD sequentially.

| Corruption | Base | Adam-NSCL | Ours | Oracle |
|---|---|---|---|---|
| gaussian_noise | 8.6 | 18.0 | 21.6 | 23.0 |
| shot_noise | 8.1 | 18.1 | 21.6 | 23.1 |
| impulse_noise | 8.7 | 18.4 | 22.1 | 23.4 |
| defocus_blur | 11.6 | 23.0 | 25.7 | 28.2 |
| glass_blur | 11.4 | 24.2 | 27.1 | 30.3 |
| motion_blur | 9.3 | 20.5 | 22.7 | 24.3 |
| zoom_blur | 2.7 | 6.2 | 6.9 | 10.7 |
| snow | 9.6 | 20.0 | 23.0 | 24.9 |
| frost | 10.1 | 20.3 | 22.9 | 24.2 |
| fog | 14.2 | 26.7 | 29.6 | 32.4 |
| brightness | 12.9 | 25.5 | 28.8 | 31.6 |
| contrast | 8.7 | 20.5 | 21.8 | 26.2 |
| elastic_transform | 12.0 | 24.9 | 28.6 | 33.3 |
| pixelate | 12.2 | 24.5 | 28.2 | 31.6 |
| jpeg_compression | 11.5 | 22.5 | 25.9 | 28.2 |
| **Average** | **10.1** | **20.8** | **23.8** | **26.4** |
| **COCO** | **59.7** | **57.2** | **57.5** | **–** |

# D ADDITIONAL BASELINE COMPARISONS AND RESOURCE EFFICIENCY

## D.1 COMPARISON WITH STATE-OF-THE-ART METHODS

To further validate the effectiveness of ABME, we included two additional state-of-the-art continual learning baselines: SGP (Saha & Roy, 2023) and SVFCL (Wang et al., 2025). We evaluated all methods under the CDFSOD 5-shot setup. As shown in Table 13, ABME significantly outperforms these recent approaches.

Table 13: Extended comparison on CDFSOD 5-shot. We incorporated two additional baselines: SGP and SVFCL. ABME achieves the best trade-off between plasticity (New-task mAP) and stability (Old-task mAP).

| Method | New-task mAP ↑ | Old-task mAP ↑ | RR ↑ | AGR ↑ |
|---|---|---|---|---|
| EWC (2017) | 31.0 | 57.1 | 95.6% | 78.3% |
| Adam-NSCL (2021) | 29.1 | 57.8 | 96.8% | 73.5% |
| SGP (2023) | 32.6 | 46.1 | 77.2% | 82.3% |
| SD-LoRA (2025) | 24.9 | 54.2 | 90.8% | 62.9% |
| SVFCL (2025) | 33.2 | 52.9 | 88.6% | 83.8% |
| **ABME (Ours)** | **38.5** | **57.0** | **95.5%** | **97.2%** |
| Oracle | 39.6 | 59.7 | – | – |

## D.2 RESOURCE CONSUMPTION ANALYSIS

We compared the training time, GPU memory usage, and extra storage cost of all methods on the CDFSOD 5-shot setup. All models were trained on the same hardware configuration. As detailed in Table 14, ABME maintains **constant resource efficiency** comparable to standard FFN fine-tuning, requiring only a fixed ~1.3 GB storage independent of the task count. In contrast, baselines exhibit varying overheads: EWC accumulates storage costs as tasks are added ($0.7 \rightarrow 3.4$ GB), while Adam-NSCL requires increased memory for subsequent tasks compared to the initial stage

$(1640 \rightarrow 1731$ MB). SD-LoRA suffers from linear memory growth and the longest training time. Furthermore, our final optimization step (solving Eq. 5) typically completes in $<$**10 seconds**, incurring negligible operational overhead.

Table 14: Resource consumption analysis on CDFSOD 5-shot. For metrics formatted as "Start $\rightarrow$ End" (e.g., 932 $\rightarrow$ 977), the values correspond to the consumption recorded during the training of the **first task** and the **final task**, respectively, indicating the variation in resource usage as tasks accumulate.

| Method | Training Time (s) | GPU Memory (MB) | Extra Storage (GB) | CDFSOD (New) | COCO (Old) |
|---|---|---|---|---|---|
| SD-LoRA | 1121 | $932 \rightarrow 977$ | – | 24.9 | 54.2 |
| EWC | 1044 | 1640 | $0.7 \rightarrow 3.4$ | 31.0 | 57.1 |
| Adam-NSCL | 904 | $1640 \rightarrow 1731$ | 1.1 | 29.1 | 57.8 |
| **ABME (Ours)** | 1029 | 1640 | 1.3 | 38.5 | 57.0 |
| Oracle (FFN-out) | 1024 | 1640 | – | 39.6 | – |

## E    CORE IMPLEMENTATION CODE

We provide the core PyTorch implementation of the Auto-Balanced Model Editing (ABME) algorithm below. For clarity, we present the solver for a single layer. The input `stats` dictionary contains the accumulated sufficiency statistics extracted from the support set:

- `kk`: The autocorrelation matrix of keys $K^\top K \in \mathbb{R}^{d_{in} \times d_{in}}$.
- `kv`: The cross-correlation matrix $K^\top V \in \mathbb{R}^{d_{in} \times d_{out}}$.
- `sum_x`, `sum_y`: Sum of input keys and output values (for bias handling).
- `n`: Total number of support samples.

```python
import torch

@torch.no_grad()
def solve_abme_layer(W_old, b_old, stats, eps=1e-8):
    """
    Apply ABME update to a single linear layer (FFN output).

    Args:
        W_old: Original weights [out_dim, in_dim]
        b_old: Original bias [out_dim]
        stats: Dictionary containing K^T K, K^T V, etc.
    """
    out_dim, in_dim = W_old.shape

    # -------------------------------------------------------
    # 1. Load Statistics
    # -------------------------------------------------------
    # kk: K^T K [in, in], kv: K^T V [in, out]
    kk = stats['kk'].to(W_old.dtype)
    kv = stats['kv'].to(W_old.dtype)
    sx = stats['sum_x'].to(W_old.dtype).reshape(in_dim, 1)
    sy = stats['sum_y'].to(W_old.dtype).reshape(1, out_dim)
    n  = torch.tensor(stats['n'], dtype=W_old.dtype)

    # -------------------------------------------------------
    # 2. Construct Augmented System (incorporating bias)
    # -------------------------------------------------------
    # A corresponds to the augmented K^T K matrix
    A = torch.zeros((in_dim + 1, in_dim + 1), dtype=W_old.dtype)
    A[:in_dim, :in_dim] = kk
    A[:in_dim, in_dim:] = sx
    A[in_dim:, :in_dim] = sx.t()
    A[in_dim, in_dim]   = n
```

```
34
35      # B corresponds to the augmented K^T V matrix
36      B = torch.zeros((in_dim + 1, out_dim), dtype=W_old.dtype)
37      B[:in_dim, :] = kv
38      B[in_dim:, :] = sy
39
40      # ----------------------------------------------------------
41      # 3. Prepare Original Parameters [W0; b0]
42      # ----------------------------------------------------------
43      W0_aug = torch.cat([W_old.t(), b_old.reshape(1, out_dim)], dim=0)
44
45      # ----------------------------------------------------------
46      # 4. Compute Auto-Balanced Regularization Matrix (Gamma^2)
47      # ----------------------------------------------------------
48      # Get diagonal energy s_i = (K^T K)_ii
49      s = kk.diag()
50
51      # Numerical stability
52      floor = s.mean() * eps
53      s = torch.clamp(s, min=floor)
54
55      # Our design: Gamma = diag(s^{1/4}) => Gamma^2 = diag(sqrt(s))
56      P = torch.zeros((in_dim + 1,), dtype=W_old.dtype)
57      P[:in_dim] = s.sqrt()       # Regularization for weights
58      P[in_dim]  = torch.sqrt(n) # Regularization for bias
59
60      # ----------------------------------------------------------
61      # 5. Solve the Linear System
62      # ----------------------------------------------------------
63      # Objective: (A + Gamma^2) W* = (B + Gamma^2 W0)
64      A_reg = A + torch.diag(P)          # LHS Matrix
65      RHS   = B + W0_aug * P.unsqueeze(1) # RHS Term
66
67      # Solve using Cholesky or LU (torch.linalg.solve)
68      W_new_aug = torch.linalg.solve(A_reg, RHS)
69
70      # ----------------------------------------------------------
71      # 6. Extract Updated Weights and Bias
72      # ----------------------------------------------------------
73      W_new = W_new_aug[:in_dim, :].t() # [out_dim, in_dim]
74      b_new = W_new_aug[in_dim, :]      # [out_dim]
75
76      return W_new, b_new
```

Listing 1: PyTorch implementation of the ABME update for a single layer.

