# OpenReview forum: "Retain and Adapt: Auto-Balanced Model Editing for Open-Vocabulary Object Detection under Domain Shifts"
_ICLR.cc/2026/Conference — ICLR 2026 Poster_

### Official Review · Reviewer_mH4j · 2025-10-22

**Soundness:** 3
**Presentation:** 3
**Contribution:** 2
**Rating:** 4
**Confidence:** 3

**Summary:**

The paper proposes ABME, which extends OVOD models to new domains without retraining. ABME injects new knowledge into FFNs through compact KV representations. Experiments on CDFSOD and ODinW-13 show that ABME maintains over 95% of base performance while achieving up to 97% adaptation gains on new domains, outperforming baselines like EWC, Adam-NSCL, and SD-LoRA.

**Strengths:**

1. ABME introduces model editing to OVOD for the first time.
2. ABME are validated across 19 datasets and multiple models (Grounding DINO, GLIP) with consistent performance gains.
3. The paper is well-written. The charts and formulas look clear and well-designed.

**Weaknesses:**

1. The auto-balancing mechanism lacks deeper theoretical analysis beyond empirical validation.
2. The approach assumes that FFN layers store the majority of knowledge, which might not hold for all architectures.
3. Detailed memory–accuracy trade-offs of KV storage are not analyzed.
4. The paper lacks broader open-world or long-term continual settings

**Questions:**

1. What are the limits of automatic balancing, and could it fail when domain shifts are extremely large or heterogeneous?
2. How would ABME perform in fully online continual learning with hundreds of sequential domains?
3. Can the KV-based editing be extended to multimodal or temporal models like video-language detection?

---

> ### Author Response · Authors · 2025-11-26
> **Author Responses to Reviewer mH4j (Part 1)**
>
> ---
>
> We would like to sincerely thank you for your efforts and valuable comments to improve our work!
>
> Below we address your concerns.
>
> ---
> **Q1. Theoretical justification for the auto-balancing mechanism.**
>
> **A1.** Thank you for highlighting the need for a deeper theoretical justification. We apologize for the oversight in the initial submission and provide the derivation below.
>
> The diagonal matrix $\Gamma$ is set as $\Gamma_{ii} = s_i^{1/4}$, with $s_i = \sum_t k_{ti}^2$ representing the **accumulated feature energy** of the input key matrix $K$. In practice, the raw feature energy $s_i$ exhibits extreme variations across dimensions (e.g., $10^4$ to $10^6$ in CDFSOD), creating severe scale imbalance. A fixed scalar $\lambda$ would be either negligible for high-energy dimensions or too restrictive for low-energy ones. To resolve this, we derived our $\Gamma$ design based on **optimization curvature analysis**:
>
> * **Data Term Curvature:** Let $M_k$ denote the magnitude scale of the input feature matrix $K$ (i.e., $\|\|K\|\| \propto M_k$). The Hessian of the data fitting term $\|\|KW - V\|\|_F^2$ with respect to $W$ is **$\mathbf{H}_{data} = 2K^TK$**. Consequently, the curvature scales quadratically with feature magnitude, i.e., **$\mathcal{O}(M_k^2)$**. This creates steep landscapes for high-energy features, causing them to dominate updates if not balanced.
> * **Adaptive Regularization:** By setting $\Gamma_{ii} = s_i^{1/4}$, the Hessian of our regularization term becomes $\mathbf{H}_{reg} = 2\Gamma^2$. Since $s_i \propto M_k^2$, our design implies that $\Gamma^2$ scales linearly with **$\mathcal{O}(M_k)$** (as $\Gamma^2 \propto \sqrt{s_i} \propto M_k$).
> * **Mechanism:** This **sub-quadratic scaling** ($\mathcal{O}(M_k)$ vs $\mathcal{O}(M_k^2)$) ensures regularization grows slower than the data term. It provides necessary constraints for stability while remaining flexible ("plastic") enough for high-energy features to adapt to new tasks.
>
> We have included this theoretical derivation in the revised supplementary material.
>
> ---
> **Q2. Validity of the assumption that FFN layers store knowledge across different architectures.**
>
> **A2.** Thank you for raising this insightful point regarding our architectural assumption.
> We acknowledge that knowledge distribution can vary across different network architectures. However, for the **Vision-Language (VL)** models focused on in this work (Grounding DINO, GLIP, CLIP), our theoretical analysis and extensive experiments demonstrate that targeting FFN layers is a **feasible and highly effective strategy** for few-shot adaptation.
>
> **1. Theoretical Motivation:**
> Recent research by Ghiasi et al. [1] indicates that in VL models, FFN neurons evolve to encode "complex abstract concepts" and "conceptual categories," specifically under **language supervision**. Unlike standalone vision models, Grounding DINO, GLIP, and CLIP inherently rely on FFNs to map visual signals to text-aligned semantic representations, making them the optimal target for model editing.
>
> **2. Empirical Verification (FFN vs. Self-Attention):**
> We compared editing FFN layers against Self-Attention (SA) layers on Grounding DINO. As shown below, FFN editing significantly outperforms SA editing on CDFSOD (5-shot) and recovers **97.2%** of the Layer Oracle performance, confirming FFNs as the more effective site for knowledge injection.
>
> | Editing Target | CDFSOD (mAP) | Layer Oracle (mAP) | % of Layer Oracle | COCO (mAP) |
> | :------------- | :----------: | :----------------: | :---------------: | :--------: |
> | Base Model     |     17.7     |         -          |         -         |    59.7    |
> | SA Layers      |     36.3     |        38.8        |       93.6%       |    57.8    |
> | FFN Layers     |     38.5     |        39.6        |       97.2%       |    57.0    |
>
> **3. Generalization to GLIP and CLIP:**
> We further validated this strategy on GLIP (detection) and CLIP (classification) by comparing our method against an Oracle (independent fine-tuning per dataset). ABME consistently recovers the majority of the Oracle performance:
>
> * **GLIP (Few-Shot Detection):** On CDFSOD (5-shot), ABME achieves **35.2 mAP**, closely approaching the Oracle's **37.2 mAP**.
> * **CLIP (Few-Shot Classification):** Across 11 datasets (including Food, MNIST, Caltech101, etc.), ABME achieves **74.11%** average accuracy compared to the Oracle's **80.43%**.
>
> These results confirm that for language-supervised architectures, FFN layers consistently serve as effective carriers of transferable visual knowledge. We have included these additional ablation studies and references in the revised supplementary material.
>
> **Reference:**
>
> [1] Ghiasi, A., et al. "What do vision transformers learn? a visual exploration." arXiv 2022.

---

> > ### Author Response · Authors · 2025-11-26
> > **Author Responses to Reviewer mH4j (Part 2)**
> >
> > ---
> > **Q3. Analysis of memory-accuracy trade-offs in KV storage.**
> >
> > **A3.** Thank you for suggesting a deeper look into the efficiency of our design.
> > We clarify that our compact KV storage is designed to be **independent of the number of tasks**, ensuring scalability **without any accuracy loss**.
> >
> > As evident in our final closed-form solution (Eq. 6), the calculation of the optimal weights $W^\star$ depends **exclusively** on the aggregated statistics $K^\top K$ and $K^\top V$ (along with the fixed priors):
> > $$
> > W^\star = (K^\top K + \Gamma^2)^{-1} (K^\top V + \Gamma^2 W_0)
> > $$
> > Since these covariance statistics are additive:
> > $$
> > K^\top K = \sum_t K_t^\top K_t, \quad K^\top V = \sum_t K_t^\top V_t
> > $$
> > storing these aggregated covariances yields **mathematically identical results** to retaining all task-specific $K$ and $V$ matrices. Therefore, our method achieves the exact same accuracy as retaining all raw statistics but with a **constant memory footprint**.
> >
> > ---
> >
> > **Q4 & Q5. Limits of automatic balancing under extreme domain shifts and performance in long-term continual learning.**
> >
> > **A4 & A5.** We appreciate the reviewers for raising these critical questions regarding the robustness and scalability of our approach. Since both questions address the performance limits of ABME under **extreme heterogeneous shifts** and **long-term sequential settings**, we address them together by conducting an additional large-scale stress test.
> >
> > **Experimental Setup for Extreme Scenarios:**
> >
> > To rigorously evaluate the limits of our method under extreme heterogeneous shifts and long-term adaptation, we designed a large-scale stress test extending the CDFSOD benchmark.
> >
> > * **Dataset Composition:** We combined the 6 domains of CDFSOD (ArTaxOr, Clipart1k, DeepFish, DIOR, NEU-DET, UODD) with the ImageCorruptions benchmark.
> > * **Extreme Shifts:** We applied **15 distinct corruption types** (covering noise, blur, weather, and digital categories) at the **maximum intensity level (Severity 5)**. This combination yields a total of 90 highly corrupted datasets (15 corruptions $\times$ 6 domains), representing severe distribution shifts.
> > * **Sequential Protocol (90 Tasks):** We performed 5-shot continual learning across all 90 tasks sequentially. As detailed in the table caption, the training follows a strict sequence: for each corruption type, the model adapts to the six domains in the fixed order of **ArTaxor $\rightarrow$ Clipart1k $\rightarrow$ DeepFish $\rightarrow$ DIOR $\rightarrow$ NEU-DET $\rightarrow$ UODD**. This results in a continuous stream of 90 adaptation steps.
> >
> > The detailed performance breakdown is presented below. "Oracle" represents the upper bound obtained by independent fine-tuning on each specific corrupted dataset.
> >
> > | Corruption        | Base | Adam-NSCL | Ours | Oracle |
> > | :---------------- | :--: | :-------: | :--: | :----: |
> > | gaussian_noise    | 8.6  |   18.0    | 21.6 |  23.0  |
> > | shot_noise        | 8.1  |   18.1    | 21.6 |  23.1  |
> > | impulse_noise     | 8.7  |   18.4    | 22.1 |  23.4  |
> > | defocus_blur      | 11.6 |   23.0    | 25.7 |  28.2  |
> > | glass_blur        | 11.4 |   24.2    | 27.1 |  30.3  |
> > | motion_blur       | 9.3  |   20.5    | 22.7 |  24.3  |
> > | zoom_blur         | 2.7  |    6.2    | 6.9  |  10.7  |
> > | snow              | 9.6  |   20.0    | 23.0 |  24.9  |
> > | frost             | 10.1 |   20.3    | 22.9 |  24.2  |
> > | fog               | 14.2 |   26.7    | 29.6 |  32.4  |
> > | brightness        | 12.9 |   25.5    | 28.8 |  31.6  |
> > | contrast          | 8.7  |   20.5    | 21.8 |  26.2  |
> > | elastic_transform | 12.0 |   24.9    | 28.6 |  33.3  |
> > | pixelate          | 12.2 |   24.5    | 28.2 |  31.6  |
> > | jpeg_compression  | 11.5 |   22.5    | 25.9 |  28.2  |
> > | Average           | 10.1 |   20.8    | 23.8 |  26.4  |
> > | COCO (Retention)  | 59.7 |   57.2    | 57.5 |   --   |
> >
> > *Table: Performance comparison of different methods under various corruptions on CDFSOD 5-shot. All values represent mean Average Precision (AP) in percentage.*
> >
> > 1.  **Robustness (Addressing Q4):** Despite the extreme domain shifts where the Base model struggles (dropping to 10.1 mAP), ABME effectively adapts to these severe corruptions, achieving 23.8 mAP. This is remarkably close to the Oracle performance (26.4 mAP), which fine-tunes independently on each task.
> > 2.  **Long-term Scalability (Addressing Q5):** Even after sequentially learning **90 distinct tasks**, ABME maintains 96.3% of the original COCO performance. This demonstrates that our automatic balancing mechanism remains stable and does not collapse or suffer from catastrophic forgetting, even in fully online settings with hundreds of potential domains.
> >
> > We have included this comprehensive analysis and the detailed experimental setup in the revised supplementary material to further demonstrate the limits and capabilities of ABME.

---

> ### Author Response · Authors · 2025-11-26
> **Author Responses to Reviewer mH4j (Part 3)**
>
> **Q6. Generalization to other multimodal tasks.**
>
> **A6.** Thank you for this insightful question regarding the extensibility of our method.
> To verify that ABME generalizes beyond open-vocabulary object detection (OVOD), we applied it to CLIP-based image classification, a fundamental multimodal task. We conducted experiments under the 5-shot setting across 11 standard datasets (including Food, MNIST, OxfordPet, Flowers, SUN397, Aircraft, Caltech101, DTD, EuroSAT, CIFAR100, and StanfordCars).
>
> We compared our ABME, which edits a *single model* with auto-balanced $\Gamma$, against FFN-out fine-tuning (Oracle) (where each dataset is independently fine-tuned) and recent methods. The results are summarized below:
>
> | Method                       | Average Accuracy (%) |
> | :--------------------------- | :------------------: |
> | ZSCL [1]                     |         67.4         |
> | WiSE-FT [2]                  |         71.9         |
> | **ABME (Ours)**              |         74.1         |
> | FFN-out Fine-tuning (Oracle) |        (80.4)        |
>
> As shown in the table, ABME outperforms continual learning baselines (ZSCL) and robust fine-tuning methods (WiSE-FT), effectively narrowing the gap with the Oracle. These results confirm that our compact KV-based editing mechanism is versatile and effective for other multimodal architectures like CLIP.
> We have added these additional experimental results and discussions to the revised supplementary material.
>
> **Reference:**
>
> [1] Zheng, Zangwei, et al. "Preventing zero-shot transfer degradation in continual learning of vision-language models." *ICCV*. 2023.
>
> [2] Wortsman, Mitchell, et al. "Robust fine-tuning of zero-shot models." *CVPR*. 2022.

---

### Official Review · Reviewer_oBvk · 2025-10-23

**Soundness:** 3
**Presentation:** 3
**Contribution:** 3
**Rating:** 6
**Confidence:** 4

**Summary:**

An innovative Auto-Balanced Model Editing (ABME) framework is proposed and applied to the domain of open-vocabulary object detection. By efficiently fine-tuning FFN layers, it achieves performance comparable to full model fine-tuning while effectively mitigating the issue of catastrophic forgetting. This research provides a novel reference solution for domain adaptation problems, demonstrating certain academic value and application potential.

**Strengths:**

1.The idea that transitioning the model editing technology from LLM to domain adaptation tasks in OVOD is motivating.

2.The proposed use of a data-driven regularization matrix to replace manual hyperparameter tuning is good.

3.The paper demonstrates clear logic, rigorous theoretical derivation.

**Weaknesses:**

1.There is no ablation on the choice of edited regions, like which layers of the FFN. To my knowledge, model editing at different layers in LLM has a significant impact.

2.The author did not provide the code for the paper, which to some extent reduces the reproducibility and practical impact of the study.

3.The baseline set is limited. The baselines in this paper , like Adam-NSCL(2021),EWC(2017) are too old and outdated.

**Questions:**

1.\begin{equation}
\min_{W} \|KW - V\|_F^2 + \|\Gamma(W - W_0)\|_F^2,
\end{equation}


\begin{equation}
\Gamma = \text{diag}\left(s_1^{1/4}, s_2^{1/4}, \ldots, s_d^{1/4}\right), \quad s_i = \sum_t k_{ti}^2.
\end{equation}
What is the rationale behind the specific definition of $\Gamma$? Why is $s_i^2$ not the  $s_i$? Is it derived from the least squares method? Have you considered the Gaussian kernel?


2.Can authors provide a comparison with more recent competitors.

---

> ### Author Response · Authors · 2025-11-26
> **Author Responses to Reviewer oBvk (Part 1)**
>
> ---
> We would like to sincerely thank you for your efforts and valuable comments to improve our work!
>
> Below we address your concerns.
>
> ---
> **Q1. Ablation study on the choice of edited layers.**
>
> **A1.** We appreciate the reviewer’s suggestion and have conducted layer-wise ablation experiments to analyze the impact of editing different regions. These experiments were performed under the CDFSOD 5-shot setting. The results are summarized in the table below, where 'New task mAP' denotes the average performance across the six novel tasks (datasets) of CDFSOD.
>
> | Edited Region (Grounding DINO, 5-shot CDFSOD) | New-task mAP $\uparrow$ | COCO AP (old) $\uparrow$ |
> | :-------------------------------------------- | :---------------------: | :----------------------: |
> | Backbone only                                 |          35.1           |           58.3           |
> | Encoder–Decoder only                          |          33.9           |           56.5           |
> | Backbone (shallow)                            |          23.4           |           56.5           |
> | Backbone (last 3 layers)                      |          20.6           |           59.3           |
> | Backbone (middle layers)                      |          22.3           |           59.6           |
> | Backbone (middle) + Encoder–Decoder           |          36.2           |           58.4           |
> | Backbone + Encoder–Decoder                    |          38.5           |           57.0           |
>
> The results indicate that our ABME strategy (editing the output layers of all FFN blocks) achieves the most favorable trade-off between new task adaptation and old knowledge retention compared to editing restricted sub-modules. We have included these ablation results in the revised supplementary material.
>
> ---
> **Q2. Concern regarding code availability and reproducibility.**
>
> **A2.** We fully agree with the importance of reproducibility.
> We are committed to open-sourcing our work and will release the complete codebase upon acceptance to maximize practical impact. To provide immediate transparency and facilitate verification, we have added Appendix E in the revised manuscript, which presents the core PyTorch implementation of the ABME algorithm. This code snippet explicitly demonstrates the closed-form update logic for FFN layers (Eq. 5) using pre-computed statistics ($K^\top K$ and $K^\top V$), verifying the method's simplicity and reproducibility.
>
> ---
>
> **Q3. Request for comparison with recent baselines.**
>
> **A3.** We sincerely thank the reviewer for this constructive suggestion to strengthen our experimental evaluation.
> Following your advice, we have expanded our baseline comparison to include two recent methods: SGP (2023) [3] and SVFCL (2025) [4], alongside the previously included SD-LoRA (2025) [5], EWC (2017) [1], and Adam-NSCL (2021) [2].
>
> The comparison is conducted under the CDFSOD 5-shot setting. As shown in the table below:
>
> * **New-task mAP** denotes the average performance across the six CDFSOD target datasets.
> * **Old-task mAP** represents performance on the COCO base domain.
>
> | Method           | New-task mAP ↑ | Old-task mAP ↑ | RR ↑  | AGR ↑ |
> | :--------------- | :------------: | :------------: | :---: | :---: |
> | EWC (2017)       |      31.0      |      57.1      | 95.6% | 78.3% |
> | Adam-NSCL (2021) |      29.1      |      57.8      | 96.8% | 73.5% |
> | SGP (2023)       |      32.6      |      46.1      | 77.2% | 82.3% |
> | SD-LoRA (2025)   |      24.9      |      54.2      | 90.8% | 62.9% |
> | SVFCL (2025)     |      33.2      |      52.9      | 88.6% | 83.8% |
> | **ABME (Ours)**  |      38.5      |      57.0      | 95.5% | 97.2% |
> | Oracle           |     (39.6)     |    (59.7)     |  --   |  --   |
>
> The results demonstrate that ABME significantly outperforms recent competitors while maintaining a superior balance between plasticity and stability (AGR of 97.2%).
>
> **References:**
>
> [1] Kirkpatrick, James, et al. *Overcoming Catastrophic Forgetting in Neural Networks.* PNAS, 2017.
>
> [2] Wang, Shipeng, et al. *Training Networks in Null Space of Feature Covariance for Continual Learning.* CVPR, 2021.
>
> [3] Saha, Gobinda, and Kaushik Roy. *Continual Learning with Scaled Gradient Projection.* AAAI, 2023.
>
> [4] Wang, Zhiwu, et al. *Singular Value Fine-tuning for Few-Shot Class-Incremental Learning.* arXiv, 2025.
>
> [5] Wu, Piao, et al. *SD-LoRA: Scalable Decoupled Low-Rank Adaptation for Class-Incremental Learning.* ICLR, 2025.

---

> > ### Author Response · Authors · 2025-11-26
> > **Author Responses to Reviewer oBvk (Part 2)**
> >
> > ---
> >
> > **Q4. Rationale behind the definition of $\Gamma$ and feasibility of using Gaussian kernels.**
> >
> > **A4.** Thank you for this insightful question regarding the theoretical grounding of our regularization design.
> >
> > **1. Rationale for $\Gamma$ and Exponent Design ($\alpha=1/4$).**
> > We define $\Gamma = \text{diag}(s_i^\alpha)$ to control regularization strength relative to feature energy. We specifically chose $\alpha=1/4$ based on a critical trade-off analysis between plasticity and stability, as opposed to the theoretical alternative $\alpha=1/2$.
> >
> > * **Theoretical Analysis (Curvature):**
> >   * **Data Term:** Let $M_k$ denote the magnitude scale of the input feature matrix $K$ (i.e., $\|\|K\|\| \propto M_k$). The Hessian of the data fitting term $\|\|KW - V\|\|\_F^2$ with respect to $W$ is **$2K^TK$**. Consequently, the curvature scales quadratically with feature magnitude, i.e., **$\mathcal{O}(M_k^2)$**. This creates steep landscapes for high-energy features, causing them to dominate updates if not balanced.
> >   * **Regularization Term:** With our design $\Gamma_{ii} = s_i^{1/4}$, the Hessian of the regularization ($2\Gamma^2$) scales linearly with $\mathcal{O}(M_k)$ (since $\Gamma^2 \propto \sqrt{s_i} \propto M_k$).
> >   * **Mechanism:** This **sub-quadratic scaling** ($\mathcal{O}(M_k)$ vs. $\mathcal{O}(M_k^2)$) ensures that regularization grows with feature strength to provide constraints, but grows *slower* than the data term to prevent over-rigidity, allowing sufficient knowledge injection for high-signal features.
> >
> > * **Empirical Validation:**
> >   As shown in the table below, $\alpha=1/2$ over-constrains the model ($\mathcal{O}(M_k^2)$ scaling matches the data term too closely), leading to lower adaptation performance. In contrast, $\alpha=1/4$ achieves a superior trade-off.
> >
> > | $\alpha$ | CDFSOD (New) | COCO (Old) | Analysis                                                     |
> > | :------- | :----------- | :--------- | :----------------------------------------------------------- |
> > | 1/2      | 36.6         | 57.0       | **Over-constrained:** Scales equalize ($\mathcal{O}(M_k^2)$), hindering adaptation. |
> > | 1/4      | 38.5         | 57.0       | **Balanced:** Sub-quadratic scaling ($\mathcal{O}(M_k)$) optimizes plasticity-stability. |
> >
> > **2. Discussion on Gaussian (RBF) Kernels.**
> > We appreciate the suggestion of using a Gaussian kernel. While theoretically sound, it is **impractical regarding storage scalability** in our incremental setting:
> >
> > * **Gaussian Kernel:** Requires storing **all original key–value pairs** ($K, V$) to compute distances. Even in a 5-shot scenario, storing features for ~50 layers across 500 samples would require **tens of gigabytes** of memory, scaling linearly with the number of samples.
> > * **Our Approach (ABME):** We only need to store **aggregated covariance statistics** ($K^\top K, K^\top V$). These are fixed-size matrices (e.g., $1024\times1024$) independent of sample count, achieving the regularization goal with constant memory usage and high efficiency.

---

### Official Review · Reviewer_6N8D · 2025-10-26

**Soundness:** 3
**Presentation:** 4
**Contribution:** 3
**Rating:** 8
**Confidence:** 4

**Summary:**

This paper presents a method for open-vocabulary object detection that reframes adaptation as a model-editing problem. Instead of standard fine-tuning or rehearsal-based continual learning, the proposed approach leverages compact key–value statistics extracted from feed-forward layers to encode new task knowledge. The method automatically balances adaptation and retention through a data-driven regularization mechanism, eliminating the need for manually tuned hyperparameters. By maintaining only aggregated statistics, the system supports scalable and order-agnostic task integration without storing original data. Empirical results on multiple few-shot detection benchmarks show that the approach achieves strong performance in new domains while effectively preserving prior knowledge. Conceptually, the work bridges ideas from knowledge editing in large language models to vision-based detectors, offering a lightweight and interpretable alternative for cross-domain adaptation.

**Strengths:**

1. Framing few-shot detection as a model-editing problem is conceptually elegant and provides a refreshing perspective that clearly distinguishes this work from conventional fine-tuning and continual learning paradigms.
2. The proposed method delivers a precise and effective solution to the central challenge of balancing adaptation and retention, offering a well-grounded formulation that is both simple and practical.
3. Extensive experiments on ODinW-13 and CDFSOD demonstrate consistent gains in adaptation performance while maintaining strong retention of prior knowledge, reinforcing the robustness and practical value of the proposed approach.

**Weaknesses:**

While the paper presents an elegant and well-motivated formulation, it inherits the core assumption from LLM-based editing that knowledge primarily resides in FFN layers. The direct application of this assumption to vision detectors may not be fully validated, as the distribution of transferable representations could differ across modalities. The paper would be strengthened by a more detailed analysis or empirical study investigating whether FFN weights indeed serve as the main locus of visual knowledge.

**Questions:**

In section 4.2, the paper replaces the pre-defined scalar regularization weight $\lambda$ with a data-adaptive diagonal matrix $\Gamma$, which scales each feature dimension according to the key-vector energy. While this design is intuitively appealing and empirically effective, the underlying rationale is only briefly mentioned, and the referred Appendix B.2 seems missing. Could the authors elaborate on the specific motivation for this formulation.

---

> ### Author Response · Authors · 2025-11-26
> **Author Responses to Reviewer 6N8D (Part 1)**
>
> ---
>
> We would like to sincerely thank you for your efforts and valuable comments to improve our work!
> Below we address your concerns.
>
> ---
> **Q1. Validity of the assumption that FFN layers serve as the primary knowledge locus in vision detectors.**
>
> **A1.** We thank the reviewer for this constructive suggestion. To rigorously address this, we combined **theoretical analysis** with **extensive ablation studies across three language-supervised architectures** (Grounding DINO, GLIP, and CLIP). The results confirm that FFNs are indeed the optimal editing locus.
>
> **1. Theoretical Evidence: FFN as the Core Locus.**
> Recent literature supports the centrality of FFNs:
>
> * **Structural:** MetaFormer [1] shows that FFNs are indispensable for feature transformation, unlike attention which acts as a token mixer.
> * **Semantic:** Ghiasi et al. [2] find that FFN neurons in Vision-Language models evolve to encode "complex abstract concepts." Since Grounding DINO, GLIP, and CLIP are all **language-supervised**, they inherently rely on FFNs to map visual signals to text-aligned semantic representations.
>
> **2. Empirical Verification: Multi-Model Ablation Study.**
> We verified the optimality of FFN layers by comparing them against Self-Attention (SA) layers and Full Fine-tuning. "Oracle" denotes the upper-bound performance of independent fine-tuning.
>
> **(1) Grounding DINO: FFN vs. Self-Attention (SA).**
> On CDFSOD (5-shot), editing FFN layers significantly outperforms editing SA layers, confirming FFNs as the more effective site for knowledge injection.
>
> | Editing Target | CDFSOD (mAP) | Layer Oracle (mAP) | % of Layer Oracle | COCO (mAP) |
> | :------------- | :----------- | :----------------- | :---------------- | :--------- |
> | Base Model     | *17.7*       | -                  | -                 | 59.7       |
> | SA Layers      | 36.3         | 38.8               | 93.6%             | 57.8       |
> | FFN Layers     | 38.5         | 39.6               | 97.2%             | 57.0       |
>
> **(2) GLIP & CLIP: FFN vs. Full Fine-tuning.**
> We further validated our approach on GLIP (CDFSOD) and CLIP (11-dataset avg. classification). FFN-based methods achieve comparable or better performance than full fine-tuning (Oracle).
>
> * **GLIP Results:**
>
> | Method                  | CDFSOD (mAP) | COCO (mAP) |
> | :---------------------- | :----------- | :--------- |
> | Base Model              | *19.1*       | 59.4       |
> | **FFN ABME (Ours)**     | 35.2         | 58.1       |
> | Full Fine-tune (Oracle) | (32.7)       | -          |
> | FFN Fine-tune (Oracle)  | (37.2)       | -          |
>
> * **CLIP Results:**
>
> | Method                  | Average Accuracy (%) |
> | :---------------------- | :------------------- |
> | Base Model              | 65.3                 |
> | **FFN ABME (Ours)**     | 74.1                 |
> | Full Fine-tune (Oracle) | (77.5)               |
> | FFN Fine-tune (Oracle)  | (80.1)               |
>
> Both theoretical and empirical results confirm that FFN layers are the effective locus for visual knowledge editing in these architectures. We have included this analysis in the revised supplementary material.
>
> **References:**
>
> [1] Yu, W., et al. "Metaformer is actually what you need for vision." CVPR 2022.
>
> [2] Ghiasi, A., et al. "What do vision transformers learn? A visual exploration." arXiv 2022.

---

> > ### Author Response · Authors · 2025-11-26
> > **Author Responses to Reviewer 6N8D (Part 2)**
> >
> > ---
> > **Q2. Rationale and motivation for the adaptive regularization matrix $\Gamma$.**
> >
> > **A2.** We thank the reviewer for pointing out this oversight. We apologize for the omission of Appendix B.2 and will ensure the full derivation is included in the revised manuscript.
> >
> > The objective function in Eq. (5) is defined as:
> > $\min\_{W} \|\|(KW - V)\|\|\_F^2 + \|\|\Gamma (W - W\_0)\|\|\_F^2,$
> > where $\Gamma_{ii} = s_i^\alpha$ (with $\alpha=1/4$) and $s_i = \sum_t k_{ti}^2$ represents the accumulated feature energy. This design addresses the extreme scale variation in feature energy $s_i$ (e.g., $10^4$ to $10^6$ in CDFSOD 5-shot) by balancing optimization curvature.
> >
> > **Theoretical Motivation: Curvature Analysis**
> > We address the optimization imbalance by analyzing the **Hessian (curvature)** of the objective function with respect to the feature magnitude:
> >
> > * **Data Term Curvature:** Let $M_k$ denote the magnitude scale of the input feature matrix $K$ (i.e., $\|K\| \propto M_k$). The Hessian of the data fitting term $\|\|KW - V\|\|\_F^2$ with respect to $W$ is explicitly given by $\mathbf{H}_{data} = 2K^TK$. Consequently, the curvature scales quadratically with the feature magnitude, i.e., $\mathcal{O}(M_k^2)$. This creates extremely steep optimization landscapes for high-energy features.
> > * **Adaptive Regularization:** By setting $\Gamma_{ii} = s_i^{1/4}$, the Hessian of our regularization term becomes $\mathbf{H}_{reg} = 2\Gamma^2$. Since $s_i$ scales with $\mathcal{O}(M_k^2)$, our design implies that $\Gamma^2$ scales linearly with $\mathcal{O}(M_k)$ (as $\Gamma^2 \propto \sqrt{s_i} \propto M_k$).
> > * **Mechanism:** This **sub-quadratic scaling** ($\mathcal{O}(M_k)$ vs $\mathcal{O}(M_k^2)$) ensures that regularization grows slower than the data term. It provides necessary constraints for stability while remaining flexible enough ("plastic") to allow high-energy features to adapt to new tasks.
> >
> > We have included this detailed derivation and analysis in the revised Appendix B.3.

---

### Official Review · Reviewer_sVhG · 2025-10-28

**Soundness:** 3
**Presentation:** 2
**Contribution:** 3
**Rating:** 6
**Confidence:** 4

**Summary:**

This paper tackles the challenge of maintaining OOD robustness in Open Vocabulary Object Detection (OVOD). The authors propose Automatically Balanced Model Editing (ABME) to efficiently integrate new task knowledge while preserving pre-trained capabilities. ABME stores compact key–value representations and automatically balances old and new knowledge without retraining. It supports order-agnostic task insertion and removal, overcoming common continual learning limitations. Experiments show that ABME achieves a better trade-off between adaptation and retention than existing methods.

**Strengths:**

1. This paper is well-structured and easy to follow.

2. The idea of model editing through compact key–value representations and automatically balancing old and new knowledge is interesting and innovative.

**Weaknesses:**

1. In Eq. (5), what is the explicit advantage of the design of $\Gamma$? How does it achieve data-adaptivity? Could the authors provide experimental evidence or theoretical insights to support this design choice?

2. From Tables 1–3 and Table 6, why do RR and AGR decrease as the number of samples (shots) increases? This trend appears to contradict the common expectation that model performance should improve with larger data sample sizes. Could the authors clarify the reason behind this behaviour?

3.  Please provide a comparison of the computational cost (e.g., memory usage, training time) of the proposed method versus existing baselines.

4. Does the proposed ABME method generalize to other models and tasks, such as image classification with CLIP? I believe that including additional experiments on such tasks could strengthen the validity and generality of the proposed approach.

5. A simple approach to balance old and new knowledge is to use residual connections of weights. Could the authors demonstrate the superiority of ABME compared to this baseline, either through empirical results or theoretical justification?

**Questions:**

See Weaknesses.

---

> ### Author Response · Authors · 2025-11-26
> **Author Responses to Reviewer sVhG (Part 1)**
>
> ------
>
> We would like to sincerely thank you for your efforts and valuable comments to improve our work!
>
> Below we address your concerns.
>
> ---
> **Q1. Rationale, data-adaptivity mechanism, and validation of the $\Gamma$ design in Eq. (5).**
>
> **A1.** We thank the reviewer for highlighting the design of $\Gamma$. The objective function in Eq. (5) is defined as:
> $\min\_{W} \|\|(KW - V)\|\|\_F^2 + \|\|\Gamma (W - W\_0)\|\|\_F^2,$ where the diagonal matrix $\Gamma$ is set as $\Gamma_{ii} = s_i^{1/4}$, with $s_i = \sum_t k_{ti}^2$ representing the accumulated feature energy. The raw feature energy $s_i$ exhibits extreme variations (e.g., $10^4$ to $10^6$), creating a severe scale imbalance where a fixed scalar $\lambda$ would be either negligible for high-energy dimensions or restrictive for low-energy ones. To resolve this, we derived our $\Gamma$ design based on curvature analysis to achieving distinct advantages:
>
> **1. Feature-wise Adaptivity via Curvature Analysis.**
> We address the optimization imbalance by analyzing the Hessian (curvature) of the objective function.
>
> * **Data Term Curvature:** The Hessian of the first term is $2K^TK$. If feature $K$ has magnitude scale $M_k$, the curvature scales quadratically as $\mathcal{O}(M_k^2)$. This creates steep landscapes for high-energy features.
> * **Adaptive Regularization:** The Hessian of our regularization term is $2\Gamma^2$. Since $s_i \sim \mathcal{O}(M_k^2)$ and we set $\Gamma_{ii} = s_i^{1/4}$, our regularization strength $\Gamma^2$ scales linearly as $\mathcal{O}(M_k)$.
> * **Mechanism:** This "sub-quadratic" scaling achieves a critical balance: it provides stronger constraints for high-energy features (**Adaptivity**) but grows slower than the data term ($\mathcal{O}(M_k)$ vs. $\mathcal{O}(M_k^2)$), ensuring the regularization does not become over-rigid (**Plasticity**).
>
> **2. Experimental Validation.**
>
> **(a) Validation of Exponent Design.** To justify our choice of $\alpha=1/4$, we conducted an ablation study comparing it against the theoretical alternative $\alpha=1/2$ (where $\Gamma = \text{diag}(s_i^\alpha)$). As shown in the table below, $\alpha=1/4$ yields a superior trade-off by maintaining sub-quadratic scaling ($\mathcal{O}(M_k)$), whereas $\alpha=1/2$ implies $\mathcal{O}(M_k^2)$, creating an over-rigid constraint that hinders adaptation performance on new tasks (CDFSOD).
>
> | $\alpha$ | CDFSOD (New) | COCO (Old) | Analysis                                                     |
> | :------- | :----------: | :--------: | :----------------------------------------------------------- |
> | 1/2      |     36.6     |    57.0    | **Over-constrained:** Scales equalize ($\mathcal{O}(M_k^2)$), hindering adaptation. |
> | 1/4      |     38.5     |    57.0    | **Balanced:** Sub-quadratic scaling ($\mathcal{O}(M_k)$) optimizes plasticity-stability. |
>
> **(b) Generalization across Task Volumes.** We evaluated stability by accumulating CDFSOD tasks from $k=1$ to $6$ and comparing our method against manually tuned scalars ($\lambda$). The results demonstrate that while fixed scalars require careful tuning to balance new and old tasks, our Auto-balanced ($\Gamma$) approach consistently matches or exceeds the performance of the best tuned $\lambda$ without any manual intervention.
>
> | Method / $\lambda$       | New Task mAP ($\uparrow$) | Old Task mAP ($\uparrow$) |
> | :----------------------- | :-----------------------: | :-----------------------: |
> | $\lambda = 1$            |           53.1            |           19.3            |
> | $\lambda = 5$            |           51.5            |           36.1            |
> | $\lambda = 10$           |           52.5            |           55.4            |
> | $\lambda = 15$           |           52.8            |           55.6            |
> | $\lambda = 20$           |           50.3            |           52.8            |
> | Auto-balanced ($\Gamma$) |           53.1            |           55.7            |

---

> > ### Author Response · Authors · 2025-11-26
> > **Author Responses to Reviewer sVhG (Part 2)**
> >
> > **Continuation of Q1 (Rationale, data-adaptivity mechanism, and validation of Eq. (5)).**
> >
> > **(c) Scalability to Large-Scale Sequential Learning (Stress Test).** To rigorously verify robustness, we conducted a stress test involving 90 sequential tasks (generated via ImageCorruptions on heterogeneous domains). Our curvature-based design maintains a stable optimization trajectory even at this scale, successfully injecting the vast majority of new knowledge (23.8 mAP vs. Oracle 26.4) while retaining 96.3% of the base model's capabilities, proving it prevents optimization from becoming rigid or unstable.
> >
> > * **Dataset Composition:** We combined the 6 domains of CDFSOD (ArTaxOr, Clipart1k, DeepFish, DIOR, NEU-DET, UODD) with the ImageCorruptions benchmark.
> > * **Extreme Shifts:** We applied **15 distinct corruption types** (covering noise, blur, weather, and digital categories) at the **maximum intensity level (Severity 5)**. This combination yields a total of 90 highly corrupted datasets (15 corruptions $\times$ 6 domains), representing severe distribution shifts.
> > * **Sequential Protocol (90 Tasks):** We performed 5-shot continual learning across all 90 tasks sequentially. As detailed in the table caption, the training follows a strict sequence: for each corruption type, the model adapts to the six domains in the fixed order of **ArTaxor $\rightarrow$ Clipart1k $\rightarrow$ DeepFish $\rightarrow$ DIOR $\rightarrow$ NEU-DET $\rightarrow$ UODD**. This results in a continuous stream of 90 adaptation steps.
> >
> > The detailed performance breakdown is presented below. "Oracle" represents the upper bound obtained by independent fine-tuning on each specific corrupted dataset.
> >
> > | Corruption        | Base | Adam-NSCL | Ours | Oracle |
> > | :---------------- | :--: | :-------: | :--: | :----: |
> > | gaussian_noise    | 8.6  |   18.0    | 21.6 |  23.0  |
> > | shot_noise        | 8.1  |   18.1    | 21.6 |  23.1  |
> > | impulse_noise     | 8.7  |   18.4    | 22.1 |  23.4  |
> > | defocus_blur      | 11.6 |   23.0    | 25.7 |  28.2  |
> > | glass_blur        | 11.4 |   24.2    | 27.1 |  30.3  |
> > | motion_blur       | 9.3  |   20.5    | 22.7 |  24.3  |
> > | zoom_blur         | 2.7  |    6.2    | 6.9  |  10.7  |
> > | snow              | 9.6  |   20.0    | 23.0 |  24.9  |
> > | frost             | 10.1 |   20.3    | 22.9 |  24.2  |
> > | fog               | 14.2 |   26.7    | 29.6 |  32.4  |
> > | brightness        | 12.9 |   25.5    | 28.8 |  31.6  |
> > | contrast          | 8.7  |   20.5    | 21.8 |  26.2  |
> > | elastic_transform | 12.0 |   24.9    | 28.6 |  33.3  |
> > | pixelate          | 12.2 |   24.5    | 28.2 |  31.6  |
> > | jpeg_compression  | 11.5 |   22.5    | 25.9 |  28.2  |
> > | Average           | 10.1 |   20.8    | 23.8 |  26.4  |
> > | COCO (Retention)  | 59.7 |   57.2    | 57.5 |   --   |
> > ---
> > **Q2. Clarification on the decreasing trend of RR and AGR as shot numbers increase.**
> >
> > **A2.** Thank you for this careful observation.
> > We clarify that RR (Retention Ratio) and AGR (Adaptation Gain Ratio) are **relative metrics**. Importantly, our method’s **absolute AP on new tasks indeed increases with larger shot numbers**, consistent with the common expectation that more data yields better performance. The decreasing trend in the ratios is due to their definitions:
> >
> > * **For AGR:** Defined as $AGR = AP_{edited}^{new}/AP_{oracle}^{new}$, the denominator (Oracle) represents the ideal upper bound of single-task fine-tuning. As the number of shots increases, the Oracle's performance improves rapidly without interference. In contrast, our method must balance multiple tasks within a single model. Consequently, while our absolute performance rises, it grows slightly slower than the Oracle due to task interference, leading to a mild decline in the relative ratio.
> > * **For RR:** Defined as $RR = AP_{edited}^{old}/ AP_{pretrained}^{old}$, the denominator represents the fixed performance of the pre-trained model. As more new data (shots) are introduced, the model faces increased pressure to balance new knowledge against old, making it harder to perfectly preserve base-domain accuracy. Thus, a slight decrease in RR is a natural result of the stability-plasticity trade-off in continual learning.

---

> ### Author Response · Authors · 2025-11-26
> **Author Responses to Reviewer sVhG (Part 3)**
>
> ---
> **Q3. Comparison of computational costs (training time, memory, and storage).**
>
> **A3.** We appreciate the suggestion. We have compared the training time, GPU memory usage, and extra storage cost of ABME against baselines on the CDFSOD 5-shot setup. In the table below, the arrow ($\to$) indicates the change in consumption from the **initial task** to the **final accumulated task**, highlighting the growth trend as tasks increase.
>
> | Method           | Training Time (s) | GPU Memory (MB) | Extra Storage (GB) | CDFSOD (New) | COCO (Old) |
> | :--------------- | :---------------: | :-------------: | :----------------: | :----------: | :--------: |
> | SD-LoRA          |       1121        |  932 $\to$ 977  |         --         |     24.9     |    54.2    |
> | EWC              |       1044        |      1640       |   0.7 $\to$ 3.4    |     31.0     |    57.1    |
> | Adam-NSCL        |        904        | 1640 $\to$ 1731 |        1.1         |     29.1     |    57.8    |
> | **ABME (Ours)**  |       1029        |      1640       |        1.3         |     38.5     |    57.0    |
> | FFN-out finetune |       1024        |      1640       |         --         |     39.6     |     --     |
>
> As shown above, ABME maintains **constant resource efficiency** comparable to standard fine-tuning. Unlike baselines which suffer from increasing memory or storage costs as tasks accumulate, ABME requires only a fixed ~1.3 GB storage regardless of task count. Additionally, our final optimization step is computationally negligible, completing in **<10 seconds**.
>
> We have included this comparison in the revised supplementary material.
>
> ---
>
> **Q4. Generalizability of ABME to other models and tasks (e.g., CLIP).**
>
> **A4.** We appreciate this valuable suggestion to broaden the scope of our method.
> To verify that ABME generalizes beyond open-vocabulary detection, we applied it to **CLIP-based image classification** under the **5-shot** setting across 11 standard benchmarks (including Food, MNIST, OxfordPet, EuroSAT, CIFAR100, etc.).
>
> We compared our **ABME**, which edits a *single model* with auto-balanced $\Gamma$, against **FFN-out Fine-tuning (Oracle)**, where each dataset is *independently fine-tuned*. As shown in the table below, ABME achieves strong performance, significantly outperforming previous methods like ZSCL [1] and WiSE-FT [2] , demonstrating its effectiveness on different architectures.
>
> | Method                       | Average Accuracy (%) |
> | :--------------------------- | :------------------: |
> | ZSCL [1]                     |         67.4         |
> | WiSE-FT [2]                  |         71.9         |
> | **ABME (Ours)**              |         74.1         |
> | FFN-out Fine-tuning (Oracle) |        (80.4)        |
>
> We have included these additional experiments and detailed analysis in the revised supplementary material.
>
> **References:**
>
> [1] Zheng, Zangwei, et al. "Preventing zero-shot transfer degradation in continual learning of vision-language models." *ICCV*. 2023.
>
> [2] Wortsman, Mitchell, et al. "Robust fine-tuning of zero-shot models." *CVPR*. 2022.

---

> > ### Author Response · Authors · 2025-11-26
> > **Author Responses to Reviewer sVhG (Part 4)**
> >
> > ---
> > **Q5. Comparison with residual weight baselines.**
> >
> > **A5. Thank you for this valuable suggestion.**
> > To demonstrate the superiority of ABME, we conducted additional experiments comparing it against a residual weight baseline defined as $W_{\text{res}}(\alpha) = W_0 + \alpha (W_{\text{new}} - W_0)$. We evaluated two variants for obtaining $W_{\text{new}}$:
> >
> > 1.  **Residual-LS**: Derived by solving the unregularized least-squares objective $\min_{W} \|\|KW - V\|\|_F^2$.
> > 2.  **Residual-FT**: Obtained by standard fine-tuning of the FFN output layer on the new task.
> >
> >
> > | Method          | $\alpha$ | CDFSOD mAP (avg) $\uparrow$ | COCO mAP (old) $\uparrow$ |
> > | :-------------- | :------: | :-------------------------: | :-----------------------: |
> > | Base            |    --    |            17.7             |           59.6            |
> > | Residual-LS     |   1.0    |             0.0             |            0.0            |
> > |                 |   0.5    |             0.3             |            0.3            |
> > |                 |   0.1    |             1.5             |            0.2            |
> > |                 |   0.01   |             6.2             |            0.0            |
> > |                 |  0.001   |            18.2             |           52.0            |
> > | Residual-FT     |   2.0    |            17.1             |           40.5            |
> > |                 |   1.0    |            22.5             |           59.0            |
> > |                 |   0.5    |            20.1             |           59.6            |
> > |                 |   0.1    |            18.2             |           59.6            |
> > | **ABME (Ours)** |   Auto   |            38.5             |           57.0            |
> >
> > **Analysis:**
> >
> > * **Residual-LS collapses:** Solving the unregularized objective on sparse few-shot data causes severe overfitting and numerical instability, often yielding 0.0 mAP and destroying pretrained weights.
> > * **Residual-FT is sensitive to $\alpha$:** It faces a manual trade-off. Large $\alpha$ hurts retention (e.g., COCO drops to 40.5), while small $\alpha$ limits adaptation.
> > * **ABME Superiority:** ABME prevents degeneracy through its data-adaptive regularization term $\|\|\Gamma(W-W_0)\|\|_F^2$. This term effectively anchors the optimization to pretrained parameters using feature-wise weighting ($\Gamma$), automatically achieving a superior balance (38.5 mAP on new tasks) compared to the best manually tuned baselines.

---

### Author Response · Authors · 2025-11-26
**Author General Responses**

We thank all reviewers for their valuable suggestions and efforts.

---

Based on your comments, we have revised our main paper and supplementary materials to further improve the work. A new version has been uploaded with all changes highlighted in blue for better visibility.

---

We are pleased that all reviewers consistently appreciate our work's **novel perspective on model editing, effective balancing mechanism, extensive experimental validation, and clear presentation**.

**1. Reviewers appreciate the novelty and refreshing perspective:**

* Editing OVOD models via compact representations and automatic balancing is interesting and innovative. (Reviewer *`sVhG`*)
* Our approach overcomes continual learning limitations by supporting order-agnostic task insertion. (Reviewer *`sVhG`*)

* Framing few-shot detection as a model-editing problem is conceptually elegant. (Reviewer *`6N8D`*)
* This refreshing perspective clearly distinguishes our work from conventional fine-tuning and continual learning paradigms. (Reviewer *`6N8D`*)
* This research provides a novel reference solution, demonstrating significant academic value and application potential. (Reviewer *`oBvk`*)
* ABME introduces model editing to OVOD for the first time. (Reviewer *`mH4j`*)

**2. Reviewers recognize the effectiveness and practicality of the proposed method (ABME):**

* The proposed method efficiently integrates new task knowledge while preserving pre-trained capabilities. (Reviewer *`sVhG`*)

* The proposed method delivers a precise solution that is both simple and practical for balancing adaptation and retention. (Reviewer *`6N8D`*)
* The system supports scalable and order-agnostic task integration without storing original data. (Reviewer *`6N8D`*)

* It achieves performance comparable to full model fine-tuning while effectively mitigating the issue of catastrophic forgetting. (Reviewer *`oBvk`*)

**3. Reviewers acknowledge the extensive experiments and strong performance:**

* Achieves a better trade-off between adaptation and retention than existing methods. (Reviewer *`sVhG`*)

* Extensive experiments on ODinW-13 and CDFSOD demonstrate consistent adaptation gains and strong retention of prior knowledge. (Reviewer *`6N8D`*)
* ABME is validated across 19 datasets and multiple models (Grounding DINO, GLIP) with consistent performance gains. (Reviewer *`mH4j`*)

**4. Reviewers commend the writing quality and theoretical rigor:**

* This paper is well-structured and easy to follow. (Reviewer *`sVhG`*)
* The paper demonstrates clear logic, rigorous theoretical derivation. (Reviewer *`oBvk`*)
* The paper is well-written. The charts and formulas look clear and well-designed. (Reviewer *`mH4j`*)

---

> ### Author Response · Authors · 2025-12-03
> **Summary of Reviewer Concerns and Our Resolutions**
>
> **Below we outline how we have addressed the core concerns from each reviewer and strengthened the work accordingly.**
>
> ---
>
> **Global Major Concerns**: Across reviewers, the main concern is gaining a clearer understanding of the motivation for the automatic balancing mechanism, as well as the rationale behind selecting FFN layers as the principal editing region. We address the automatic balancing mechanism through a curvature-based analysis that explains the adaptive behavior of $\Gamma$, experiments showing its advantage over manual hyperparameter tuning, demonstrations of stable balancing across 90 long-sequence tasks, and generalization results on Grounding DINO, GLIP, and CLIP. We further support FFN editing with extensive ablations comparing FFN and self-attention editing regions, evaluations of different FFN-layer selection strategies, and theoretical insight showing that vision FFNs naturally encode text-aligned semantic representations in language-supervised models.
>
>
>
> In addition to this shared theme, individual reviewers raise some specific concerns, which we summarize below.
>
> The reviewer *`sVhG` (Rating: 6, Confidence: 4 )* expresses concerns about the decreasing trend of RR and AGR as the number of shots increases. This arises from a misunderstanding: RR and AGR are relative metrics that naturally decline as the model handles more information (shots). This is a normal phenomenon reflecting the stability-plasticity trade-off, a trend also consistently observed in baseline methods. The reviewer also requests a comparison of computational costs; we provide a detailed analysis showing that our approach matches or improves the efficiency of existing baselines without introducing additional resource overhead.
>
> The reviewer *`6N8D` (Rating: 8, Confidence: 4 )* raises concerns fully aligned with the Global Major Concern, which we address through the provided theoretical and empirical analyses.
>
> The reviewer *`oBvk` (Rating: 6, Confidence: 4 )* raises concerns about comparisons with more recent methods and alternative editing designs; we address this by adding evaluations against SGP (2023) and SVFCL (2025) alongside prior baselines, showing clear gains in adaptation (+5.3 mAP) and retention (95.5%), and by explaining that our covariance-based formulation offers fixed, task-independent storage whereas Gaussian-kernel alternatives require storing all intermediate features.
>
> The reviewer *`mH4j` (Rating: 4, Confidence: 3 )* questions the scalability of our method to longer task sequences and extreme domain shifts. To address this, beyond the original 19 cross-domain datasets (CDFSOD + ODinW), we construct a large-scale stress test combining CDFSOD with ImageCorruptions, creating 90 sequential tasks with extreme shifts to validate our method's effectiveness. Regarding the extension to other multimodal tasks, we provided additional results on CLIP for classification alongside Grounding DINO and GLIP for detection. Finally, regarding the memory-accuracy trade-offs of KV storage, we clarify that storing covariance statistics is mathematically equivalent to retaining all raw features, implying no accuracy trade-off exists. We believe we have fully addressed all the reviewer's concerns.

---

> > ### Author Response · Authors · 2025-12-03
> > **Detailed Summary of Reviewer Concerns and Our Resolutions**
> >
> > We sincerely thank all reviewers for their constructive suggestions regarding the motivation of the automatic balancing mechanism and the rationale behind selecting FFN layers. Below, we provide our responses to these **shared concerns**.
> >
> > * **Concern about the motivation of the automatic balancing mechanism:** We address this through a rigorous curvature-based analysis that explains the adaptive behavior of $\Gamma$. This analysis demonstrates how our mechanism prevents over-rigidity compared to fixed scalars. We further validate this design with experiments showing its clear advantage over manual hyperparameter tuning, demonstrations of exceptional stability across 90 long-sequence tasks under extreme domain shifts, and strong generalization results on diverse architectures including Grounding DINO, GLIP, and CLIP.
> >
> >
> > * **Concern about the rationale behind selecting FFN layers as the principal editing region:** We justify the selection of FFN layers through both empirical verification and theoretical grounding. We support FFN editing with extensive ablations comparing FFN versus Self-Attention editing regions and evaluations of different FFN-layer selection strategies. Additionally, we provide theoretical validation showing that FFNs in language-supervised vision models naturally encode text-aligned semantic representations, making them the effective locus for model editing.
> >
> > In addition to this shared theme, individual reviewers raise several more specific concerns, which we summarize below.
> >
> > **1.** We thank Reviewer *`sVhG`* for the helpful feedback. Below, we address the specific concerns regarding metric trends and computational costs.
> >
> > * **Concern about the decreasing trend of RR and AGR as shot numbers increase:** We appreciate this careful observation and clarify that this trend arises from the definition of these *relative metrics* rather than a drop in absolute performance. Our method's absolute AP indeed increases with more shots, consistent with the expectation that more data yields better performance. However, the AGR decreases because the denominator (Oracle) improves rapidly without interference, while the RR decreases slightly because the model faces increased pressure to balance new knowledge against the fixed pre-trained baseline, reflecting the natural stability-plasticity trade-off.
> >
> > * **Request for a comparison of computational costs:** We appreciate the suggestion and provide a detailed analysis comparing training time, GPU memory, and storage against baselines like EWC and Adam-NSCL. The results demonstrate that ABME maintains constant resource efficiency. Unlike some baselines where costs grow with task accumulation, ABME requires fixed storage (~1.3 GB) regardless of task count and requires negligible time (<10 seconds) for the final optimization step.
> >
> > **2.** We thank Reviewer *`6N8D`* for the high praise and positive assessment of our work’s novelty and effectiveness. Because the reviewer’s primary questions concerning the theoretical motivation of the adaptive regularization matrix $\Gamma$ and the validity of the FFN assumption represent key themes shared across reviewers, we prioritize addressing them in the shared concerns above.

---

> > > ### Author Response · Authors · 2025-12-03
> > > **Detailed Summary of Reviewer Concerns and Our Resolutions (continued)**
> > >
> > > **3.** We thank Reviewer *`oBvk`* for the constructive feedback. Below, we address the specific concerns regarding comparisons and alternative designs.
> > >
> > > * **Concern about comparisons with recent baselines:** We address this by significantly expanding our evaluation to include recent methods SGP (2023) and SVFCL (2025), alongside our previously included baselines (SD-LoRA, EWC, and Adam-NSCL). The results demonstrate that our method consistently outperforms this comprehensive set of competitors, showing clear gains in adaptation (+5.3 mAP) compared to the strongest baselines and maintaining retention (95.5%).
> > >
> > > * **Concern about alternative editing designs (e.g., Gaussian kernels):** We clarify the distinction between our approach and kernel-based alternatives regarding resource efficiency. We explain that our covariance-based formulation offers fixed, task-independent storage (aggregated statistics), whereas Gaussian-kernel alternatives require storing all intermediate features to compute distances, making them impractical for incremental settings.
> > >
> > > * **Concern about code availability and reproducibility:** We fully agree with the importance of reproducibility. To facilitate immediate verification, we add Appendix E to the revised manuscript, providing the core PyTorch implementation of the ABME algorithm (specifically the closed-form update logic). Furthermore, we are committed to releasing the complete codebase to maximize practical impact.
> > >
> > >
> > >
> > > **4.** We thank Reviewer *`mH4j`* for the valuable feedback. Below, we address the concerns regarding scalability, multimodal extension, and storage trade-offs.
> > >
> > > * **Concern about scalability to longer task sequences and extreme domain shifts:** To address this, we expand our evaluation beyond the original 19 datasets (CDFSOD + ODinW). We construct a large-scale stress test combining CDFSOD with ImageCorruptions, creating 90 sequential tasks characterized by extreme heterogeneous shifts. The results validate our method's exceptional stability in extreme long-term scenarios.
> > >
> > > * **Question about extension to other multimodal tasks:** We validate our approach's versatility by providing additional results on CLIP for classification, alongside Grounding DINO and GLIP for detection. Our method consistently narrows the performance gap with full fine-tuning oracles across these diverse architectures.
> > >
> > > * **Concern regarding memory-accuracy trade-offs of KV storage:** We clarify that our compact covariance storage involves no accuracy trade-off. Since storing aggregated covariance statistics is mathematically equivalent to retaining all raw features, our method achieves the exact same accuracy as full data retention but with a constant, task-independent memory footprint.

---

### Meta-Review · Area_Chair_241D · 2025-12-22

**Summary:**

The paper received four reviews with mostly positive scores: 6, 8, 6, and 4. Below the AC discusses the main concerns from these reviewers and how the concerns are addressed.

Reviewer sVhG (score: 6) requested a detailed comparison on the computational cost between different methods, as well as an additional baseline based on residual connections and experiments on image classification using CLIP. For computational cost, the rebuttal included training time, GPU memory, storage, and other relevant metrics in the comparison. The results show that the proposed approach is comparable to regular fine-tuning. The rebuttal also provided additional results using CLIP for image classification and the mentioned baseline. Overall the new results confirm the effectiveness of the method. The AC believes the concerns from this reviewer have been addressed. The reviewer is likely to raise the score from 6 to 8.

Reviewer 6N8D (score: 8) requested more (empirical) evidence to justify that FFN weights contain the most knowledge and editing these layers is sufficient. The rebuttal provided the requested experiment involving different editing strategies (self-attention vs. FFN), and the results confirm the assumption. This reviewer would maintain the score of 8.

Reviewer oBvk (score: 6) shared a similar concern with 6N8D and requested an ablation study on which FFN weights to edit. This reviewer also asked for more recent baselines to be compared in the paper. The rebuttal provided the ablation study and included methods developed in 2025 as the baselines. The rebuttal is clear and solid. The AC believes the concerns from this reviewer should have been fully addressed. The reviewer may maintain the score of 6 or raise it to 8.

Reviewer mH4j (score: 4) also questioned the assumption that FFN layers contain the most knowledge. The reviewer also requested more results covering broader tasks. The rebuttal addressed all of these concerns with solid experiments. The AC believes the reviewer is likely to raise the score from 4 to 6.

The final predicted scores are 6/8, 8, 6/8, and 6.

**Reviewer Concerns:**

As discussed above, the rebuttal is clear and solid. All of the reviewers' concerns should have been addressed.

**Reviewer Scores:**

The initial scores are mostly positive. The positive reviewers would maintain or increase the score. The only one negative reviewer would increase the score to 6 as his/her concerns have been addressed in the rebuttal. The AC predicts that the final scores of this work would be 6/8, 8, 6/8, and 6.

---

### Decision · Program_Chairs · 2026-01-26

Accept (Poster)